# Efficacy of anti-CD147 chimeric antigen receptors targeting hepatocellular carcinoma

Hsiang-chi Tseng [1,9], Wei Xiong[1,2,9], Saiaditya Badeti[1,9], Yan Yang [1,9], Minh Ma[1], Ting Liu[1], Carlos A. Ramos [3], Gianpietro Dotti [4], Luke Fritzky[5], Jie-gen Jiang[1], Qing Yi[2], James Guarrera[6], Wei-Xing Zong[7], Chen Liu[1] & Dongfang Liu [1,8 ✉]

Chimeric antigen receptor (CAR) therapy is a promising immunotherapeutic strategy for treating multiple refractory blood cancers, but further advances are required for solid tumor CAR therapy. One challenge is identifying a safe and effective tumor antigen. Here, we devise a strategy for targeting hepatocellular carcinoma (HCC, one of the deadliest malignancies). We report that T and NK cells transduced with a CAR that recognizes the surface marker, CD147, also known as Basigin, can effectively kill various malignant HCC cell lines in vitro, and HCC tumors in xenograft and patient-derived xenograft mouse models. To minimize any on-target/off-tumor toxicity, we use logic-gated (log) GPC3–synNotch-inducible CD147-CAR to target HCC. LogCD147-CAR selectively kills dual antigen (GPC3$^+$CD147$^+$), but not single antigen (GPC3$^-$CD147$^+$) positive HCC cells and does not cause severe on-target/off-tumor toxicity in a human CD147 transgenic mouse model. In conclusion, these findings support the therapeutic potential of CD147-CAR-modified immune cells for HCC patients.

[1] Department of Pathology, Immunology and Laboratory Medicine, Rutgers University-New Jersey Medical School, 185 South Orange Avenue, Newark, NJ 07103, USA. [2] Center for Translational Research in Hematologic Malignancies, Houston Methodist Cancer Center, Houston Methodist Research Institute, 6550 Fannin Street, SM8026, Houston, TX 77030, USA. [3] Department of Medicine, Baylor College of Medicine, One Baylor Plaza, Houston, TX 77030, USA. [4] Department of Microbiology and Immunology and Lineberger Comprehensive Cancer Center, University of North Carolina, Chapel Hill, NC 27599, USA. [5] Imaging core facility, Rutgers University-New Jersey Medical School, 205 South Orange Avenue, Newark, NJ 07103, USA. [6] Department of Surgery, New Jersey Medical School, Rutgers-The State University of New Jersey, 185 South Orange Avenue, Newark, NJ 07101, USA. [7] School of Pharmacy, Rutgers-The State University of New Jersey, Newark164 Frelinghuysen Road Piscataway, NJ 08854, USA. [8] Center for Immunity and Inflammation, New Jersey Medical School, Rutgers-The State University of New Jersey, 185 South Orange Avenue, Newark, NJ 07101, USA. [9] These authors contributed equally: Hsiang-chi Tseng, Wei Xiong, Saiaditya Badeti, Yan Yang. ✉email: dongfang.liu@rutgers.edu

Despite recent advances in chimeric antigen receptor (CAR)-T cell immunotherapy in blood cancers, high costs, complex manufacturing processes, and severe toxicity have hindered its widespread use[1]. Meanwhile, CAR-T cells face additional challenges during the targeting of solid tumors, such as identification of a safe and effective solid tumor antigen and maintenance of durable CAR cell proliferation and persistence in the tumor microenvironment[2,3]. Among the various solid tumors in humans, liver cancer is one of the deadliest tumors.

Liver cancer is the second most common cause of cancer-related death worldwide[4]. The burden of liver cancer is projected to be over 1 million cases by 2030[5]. Liver cancer ranks fifth in terms of global cases and second in terms of deaths for males[6]. More than half a million patients die from hepatocellular carcinoma (HCC) each year[7].

Primary liver cancer comprises of HCC, intrahepatic cholangiocarcinoma (iCCA), fibrolamellar carcinoma, and hepatoblastoma[4]. HCC and iCCA are the most common primary liver cancers, which account for >99% of primary liver cancer cases[4]. There are nearly 800,000 new HCC cases per year and HCC alone accounts for 90% of all cases of primary liver cancer[8].

Although liver cancer can be classified into several different types, HCC is the most common type of primary liver cancer in adults[9]. HCC is related to chronic viral hepatitis infection (including hepatitis B and C) and toxic exposure (e.g., aflatoxin B1)[10]. Currently, there is no effective therapy available to treat HCC. Sorafenib (CheckMate-040, a multikinase inhibitor) is a widely used first-line standard systemic agent for advanced HCC[11], but has been shown to have low efficacy and severe side effects[12–19]. Currently, PD-1 blockade Opdivo (Nivolumab) has been approved by the US Food and Drug Administration as a second-line treatment strategy for patients with HCC who have been previously treated with Sorafenib[20]. Clinical trials testing PD-1 blockade as a first-line treatment for HCC are underway[21]. Meanwhile, various clinical trials using PD-1 or PD-L1 blockades in combination with other interventions are ongoing as well. For example, a current study is evaluating an anti-PD-1 antibody in combination with an anti-CTLA-4 antibody in patients with resectable and potentially resectable HCC (NCT03222076).

CAR-T cell therapy has become a promising immunotherapeutic strategy for the treatment of various blood cancers[3]. CAR-modified immune cells for liver cancer therapy are rapidly being developed in both academic settings and in industry[22,23]. One remaining challenge for CAR-mediated immunotherapy for liver cancer (one of the deadliest solid tumors in humans) is to find a valid, safe, and effective target.

Current adoptive CAR-modified immune cell therapies for HCC mainly target three antigens: glypican-3 (GPC3, NCT02723942, NCT02905188, NCT02395250, NCT03146234, NCT03084380, NCT02715362, NCT03198546, NCT03302403), α-fetoprotein (AFP, NCT03349255), and mucin-1 (MUC1, NCT02587689).

In addition to GPC3-CAR[24,25], AFP-CAR[26], and MUC1-CAR[27,28] for HCC immunotherapy, adoptive TCR-T cells have also been investigated in the treatment of HCC[29,30], which includes HBV-specific TCR in HCC patients with HBV infection (NCT02686372, NCT02719782) and AFP-specific TCR (NCT03132792) in advanced HCC patients.

Dual-targeting CAR-T cells coexpressing GPC3 and asialoglycoprotein receptor 1 (ASGR1, a liver tissue-specific protein) targeting CAR's featuring CD3 zeta chain (CD3ζ) and 28BB (containing both CD28 and 4-1BB signaling domains) have been tested in a GPC3+ASGR1+ HCC tumor xenograft mouse model[31].

CD147 is a transmembrane glycoprotein with multiple functions in immune cell biology and diseases[32]. The immunological role of CD147 in immune cells is important for T cell activation and proliferation, as well as cell migration, adhesion, and invasion[32]. CD147 is expressed on different cell types (e.g., hematopoietic, epithelial, and endothelial cells) at varying levels[33]. Normal epithelial and fetal tissues have low expression of CD147, when measured by immunohistochemical analysis[34]. However, CD147 is significantly upregulated in aggressive disease states, such as in HCC[35,36]. Different groups have observed CD147 upregulation in various tumors[37,38], including breast cancer, bladder cancer, colorectal cancer, ovarian cancer, melanoma, and osteosarcoma. In the past few decades, the role of CD147 has been extensively studied in tumor biology with a focus on tumor cell proliferation, invasiveness, and metastasis[39,40]. The proposed molecular mechanism of CD147 in cancer is that CD147 on the surface of tumor cells can trigger the production of matrix metalloproteinase (a family of secreted, zinc-dependent endopeptidases capable of degrading extracellular matrix components), which facilitates tumor invasion, growth, and metastasis[41–43].

Previous studies show that CD147 overexpression may be associated with tumor cell migration and activation of the extracellular-signal-regulated kinase signaling pathway[44]. CD147 has also been proposed as a novel prognostic biomarker for HCC[36]. Specifically, development of a $^{131}$I-labeled HAb18 F(ab′)2 (Licartin) has been reported as a potential treatment option for liver cancer. Clinical trial results show that Licartin is safe and effective in patients with primary HCC[45].

Preclinical studies conducted in China show that metuzumab (anti-CD147) for the treatment of non-small cell lung cancer is safe. Metuzumab exhibits promising efficacy in inhibiting tumor growth in mouse xenograft models[46].

Despite recent advances in CAR-T cell immunotherapy, high costs and severe toxicity have hindered its widespread use[3]. To alleviate these disadvantages of CAR-T cell immunotherapy, additional cytotoxic cell-mediated immunotherapies are urgently needed. The unique biology of natural killer (NK) cells allows them to serve as a safe, effective, alternative, and possibly superior immunotherapeutic strategy to CAR-T cells, clinically[47]. Therefore, in addition to T cell-mediated immunotherapy, CAR-NK cells may serve as a valid tool for immunotherapy[48].

In this study, we hypothesize that immune cells, including T and NK cells, expressing a CD147-targeting CAR can mount a sustained anti-CD147+ HCC immune response. The biological properties of the CD147 antigen allow CD147-CAR-modified immune cells, including primary CD8+ T, primary NK, and NK-92MI cells, to produce potent anti-tumor activity against HCC in vitro and in vivo. Moreover, a synthetic Notch (synNotch) receptor "logic-gated" CD147-CAR targeting both GPC3 and CD147 is developed for minimizing potential toxicity in a human CD147 transgenic (hCD147TG) mouse model. Together, the data support the clinical development of CD147-CAR immune cell therapy for various CD147+ solid tumors.

## Results

**CD147 is expressed in HCC cell lines and patient specimens.** To determine whether CD147 is an effective, valid target for HCC and other types of cancers, we analyzed the correlation between patient survival and expression level of CD147 from TCGA (The Cancer Genome Atlas, https://cancergenome.nih.gov) datasets. Comparison of survival percentages from two different patient subsets (CD147high and CD147low) showed that there was a strong negative correlation between CD147 expression and overall survival (Supplementary Fig. 1a). Specifically, CD147high

in multiple tumor tissues were correlated with a lower survival percentage. In addition, comparisons of CD147 expression between normal tissues (NTs) and tumor patient (TP) samples across multiple cancer types showed significant upregulations of CD147 expression among different types of tumor tissues (Supplementary Fig. 1b).

To verify the data from the bioinformatics analysis, we examined the expression of CD147 among different tumor cell lines and other tissues by Western blot (WB), including wild-type NK-92 (a human NK cell line[49]), T2 (a mutant T × B cell hybrid[50]), 721.221 (a human leukocyte antigen (HLA)-A, -B, -C-null human cell line[51,52]), MDA-MB-231 (a human breast carcinoma cell line[53]), K562 (a human myelogenous leukemia cell line[54]), HepG2 (a human HCC cell line[55]), SK-Hep1 (a human liver adenocarcinoma cell line[56]), Raji (a human B lymphocyte, Burkitt's lymphoma cell line[57]), Daudi (a human B lymphoblast cell line[58]), NK-92MI (an interleukin-2 (IL-2)-independent natural killer cell line[59]), and human peripheral blood mononuclear cells (PBMCs). CD147 molecules were highly upregulated in HepG2 and SK-Hep1 cell lines (two commonly used HCC cell lines), compared to PBMCs (Supplementary Fig. 1c). The expression of CD147 on PBMCs is relatively low, compared to tumor cell lines (Supplementary Fig. 1c). Similar results were obtained by flow cytometry analysis (Supplementary Fig. 1d). Furthermore, the results of immunohistochemistry (IHC) assays confirmed that CD147 was significantly upregulated in human HCC tissue isolated from a PDX mouse model (Supplementary Fig. 2). Therefore, CD147 molecules could represent an effective, valid target for HCC treatment.

**Generation and phenotype of CD147-CAR-NK cells**. To test the susceptibility of HCC to CD147-CAR, we generated the CD147-CAR using the SFG retroviral vector[60,61]. The CD147-CAR contains the single-chain variable fragment (scFv) of the anti-CD147 antibody (derived from clone 5F6 with optimization), a human IgG1-CH2CH3 spacer, transmembrane domain of CD28, the intracellular domain of CD28-4-1BB, and intracellular signaling domains of the TCR-ζ chain (Supplementary Fig. 3a). First, we tested this CAR construct using NK-92MI cells[49,59,62]. After transduction, NK-92MI cells expressed the CD147-CAR molecules (Supplementary Fig. 3b). After sorting by flow cytometry, the percentage of CD147-postive NK-92MI cells was >96% (Supplementary Fig. 3b). We further verified the expression of CD147-CAR molecules in NK-92MI cells by WB. Compared to the parental NK-92MI cell, the CD147-CAR-NK-92MI expressed the chimeric anti-CD147-scFv-CAR. The approximate molecular weight was ~80 to 85 kDa (Supplementary Fig. 3c). We further characterized the surface immunophenotype in CD147-CAR-NK-92MI cells. Comparable activating receptor (e.g., CD56, NKG2D, NKP46, CD16, CD94/NKG2C, CD226 [also known as DNAM-1], and CD244 [also known as 2B4]) and inhibitory receptor (e.g., KLRG1, LAG-3, CTLA-4, TIM-3, PD-1, NKG2A, and TIGIT) expressions were observed (Supplementary Fig. 3d). CD147-CAR expression was associated with loss of CD147 or decreased CD147 expression on CD147-CAR-NK-92MI cells, indicating limited fratricide among CD147-CAR-NK-92MI cells (Supplementary Fig. 4). Notably, the loss of CD147 molecule expression on CD147-CAR-NK-92MI cells did not affect their functionalities and expansion in vitro. Thus, we successfully established a CD147-CAR-NK-92MI cell line for further study.

**CD147$^+$ HCC cell lines activate CD147-CAR-NK cells**. After successful establishment of CD147-CAR-NK cells, we then evaluated the capacity of CD147-CAR-NK-92MI cells to eradicate CD147$^+$ HCC cell lines (including SK-Hep1 and HepG2 cells).

Compared with control CD19-CD28-CAR-NK-92MI and CD19-CD28-4-1BB-CAR-NK-92MI cells, as described previously[61], CD147-CAR-NK-92MI cells demonstrated significant cytotoxicity against several HCC cell lines, including SK-Hep1 and HepG2 (Fig. 1a, b), as well as Huh7 and HCO2 cell lines compared to parental NK-92MI cells (Supplementary Fig. 5).

The production of tumor necrosis factor alpha (TNF-α) and interferon-γ (IFNγ) were greatly induced in CD147-CAR-NK-92MI cells, comparing to CD19-CD28-CAR-NK-92MI and CD19-CD28-4-1BB-CAR-NK-92MI cells stimulated by both SK-Hep1 and HepG2 cells (Fig. 1c, d). Interestingly, activation of CD147-CAR-NK-92MI cells by their susceptible target cells can be blocked by the anti-CD147 antibody (clone HIM6), but not the isotype IgG1 (Fig. 1e–g). The specificity of this anti-CD147 antibody was further verified by testing its effects on cytotoxicity of CD19-CD28-4-1BB-CAR-NK-92MI cells. We showed that this anti-CD147 antibody could not block the cytotoxicity of CD19-CD28-4-1BB-CAR-NK-92MI cells against CD19$^+$ Daudi cell line (Supplementary Fig. 6), indicating that the anti-CD147 antibody (clone HIM6) did not affect other types of CAR-NK cells (e.g., CD19-CD28-4-1BB-CAR cells), but specifically competed with CD147-CAR to recognize the target tumor antigen.

To further confirm the specificity of CD147-CAR-NK-92MI cells, we generated the CD147-knockout (CD147$^{-/-}$) SK-Hep1 cell line (CD147$^{-/-}$-SK-Hep1; Supplementary Fig. 7) and CD147$^{-/-}$ HepG2 cell line (CD147$^{-/-}$-HepG2; Supplementary Fig. 7). The CD147$^{-/-}$-HepG2 and CD147$^{-/-}$-SK-Hep1 cells were not recognized by CD147-CAR-NK-92MI cells (Fig. 1h–k), which was quantified by CD107a surface expression when cocultured with CD147$^{-/-}$ cell lines. Thus, we concluded that only CD147$^+$ HCC cell lines could selectively activate the CD147-CAR-NK-92MI cells.

**CD147-CAR-modified cells specifically kill HCC in vitro**. Both CD107a assay and cytokine production assay can be used to evaluate the activation of CD147-CAR-NK-92MI cells by the susceptible target cells. To directly test whether CD147-CAR-modified immune cells can kill CD147$^+$ HCC cells in vitro, we used the 4-h Chromium-51 ($^{51}$Cr) release assay, a gold standard assay for evaluating the cytotoxicity of CTLs and NK cells in the field of immunology, in combination with an FFLuc reporter assay. The FFLuc reporter assay is a non-radioactive approach for the assessment of CD147-CAR-modified immune cell cytotoxicity using a luciferase bioluminescent signal.

CD147-CAR-modified primary T and NK cells isolated from human PBMCs can kill multiple HCC cell lines (including SK-Hep1, Huh7, and HepG2, etc.), but not κ-CAR-modified T cells (Fig. 2a–c), by FFLuc reporter assays. We also demonstrated that CD147-CAR-modified human primary NK cells effectively killed HCC cell lines in vitro, by $^{51}$Cr release assay (Fig. 2d–f). To further demonstrate that the natural killing ability of primary NK through the NKG2D and NKG2D-Ligand (NKG2D-L) interaction in addition to CD147-CAR primary NK cytotoxicity, we observed that anti-NKG2D further blocked the killing of CD147$^{-/-}$-SK-Hep1 cells by CD147-CAR-NK (Fig. 2g).

The dose-dependent specific lysis of SK-Hep1 (Fig. 2i) using the FFLuc reporter assay is comparable with $^{51}$Cr release assays (Fig. 2h). The cytotoxicity by these two complementary approaches was further directly quantified under a common inverted fluorescence microscope to evaluate the morphology and dynamics of GFP signal in target cells (Supplementary Fig. 8 and Supplementary Movie 1). Together, these complementary approaches demonstrate that CD147-CAR-NK cells specifically kill CD147$^+$ HCC in vitro.

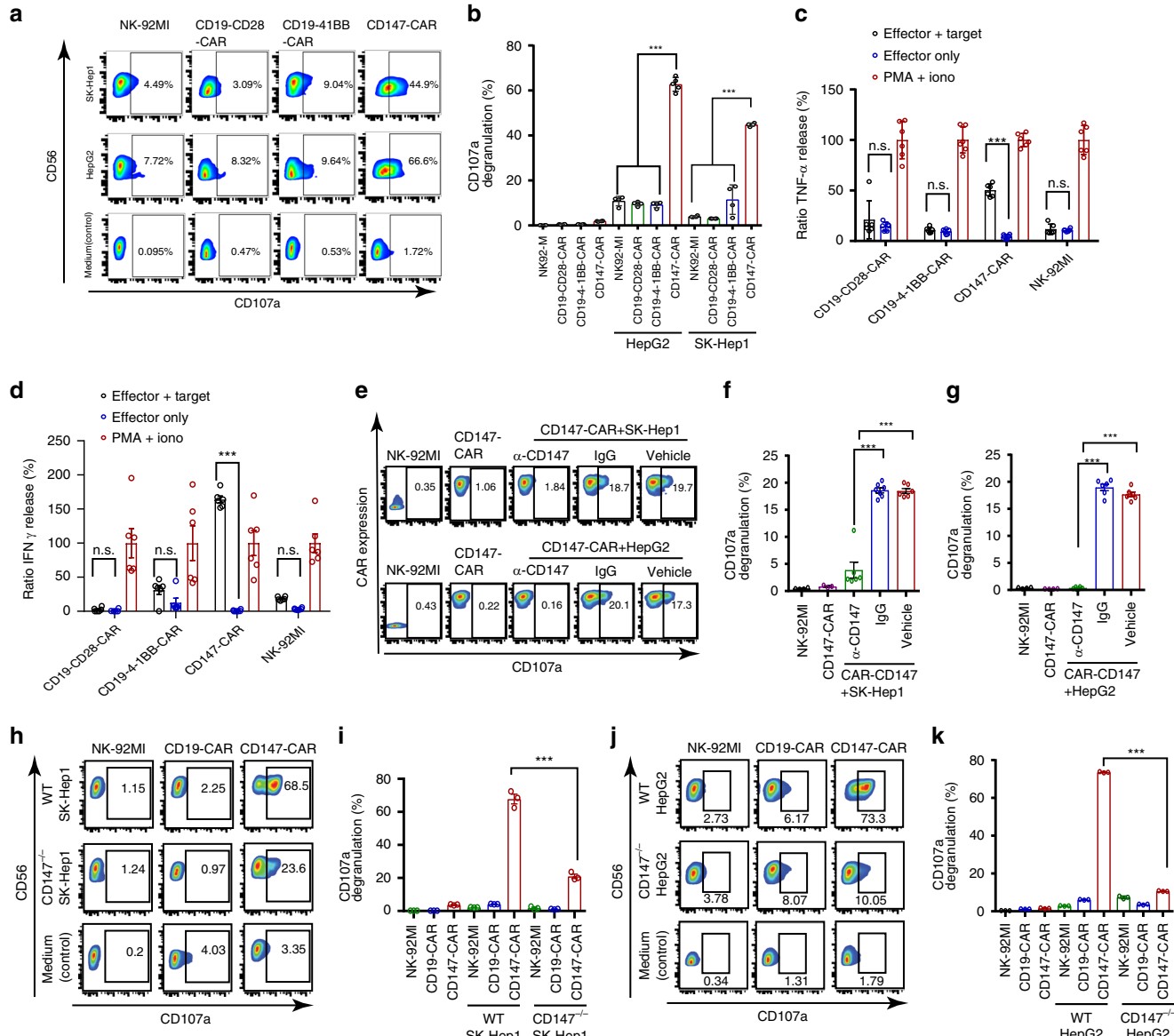

**Fig. 1 Activation profiles of CD147-CAR-NK-92MI stimulated with sensitive target cells. a** Representative flow cytometric data illustrating CD107a degranulation by NK-92MI, CD19-4-1BB-CAR, CD19-CD28-CAR, and CD147-CAR after mixing with the medium (control), SK-Hep1, and HepG2. **b** Quantitative data for the percentage of surface CD107a expression on CD147-CAR-NK-92MI cells upon different stimulations, as indicated. Data represent the mean ± SEM ($n = 4$). **c**, **d** Cytokine TNF-α (**c**) and IFNγ (**d**) production by CD147-CAR-NK-92MI, CD19-4-1BB-CAR-NK-92MI, CD19-CD28-CAR-NK-92MI, and wild-type NK-92MI stimulated by different conditions. Data are pooled from four independent experiments. **e** Representative data showing the percentage of surface CD107a expression on CD147-CAR-NK-92MI cells upon different stimulations, as indicated. Isotype-mouse IgG (IgG) or PBS (vehicle control group) was used, as indicated. **f**, **g** Quantitative data for the percentage of surface CD107a staining on CD147-CAR-NK-92MI cells stimulated with CD147+ SK-Hep1 and CD147+ HepG2 cell lines. Wild-type NK-92MI cells alone and CD147-CAR-NK-92MI cells alone were used as controls, as indicated. Data represent the mean ± SEM of four independent experiments. **h** Representative data showing the percentage of surface CD107a expression on CD147-CAR-NK-92MI cells stimulated with CD147+ wild-type (WT) SK-Hep1 cell line (top panel) and CD147-knockout (CD147−/−) SK-Hep1 cell line (middle panel). **i** Quantitative data for the percentage of surface CD107a staining on CD147-CAR-NK-92MI cells stimulated with CD147+ SK-Hep1 (WT) and CD147-knockout SK-Hep1 (CD147−/−) cell lines, respectively. Data represent the mean ± SEM of three independent experiments. **j** Representative data showing the percentage of surface CD107a expression on CD147-CAR-NK-92MI cells stimulated with a CD147+ wild-type (WT) HepG2 cell line (top panel) and a CD147-knockout HepG2 cell line (middle panel). **k** Quantitative data for the percentage of surface CD107a staining on CD147-CAR-NK-92MI cells stimulated with CD147+ HepG2 (WT) and CD147-knockout HepG2 (CD147−/−) cell lines, respectively. Data represent the mean ± SEM of three separate experiments. Unpaired Student's $t$ test was employed for all the panels. ***$p < 0.001$; n.s. not significant.

Expectedly, CD147-CAR-NK-92MI cells could not kill CD147−/−-SK-Hep1, and CD147−/−-HepG2 cells, compared to parental SK-Hep1 and HepG2 cells (Fig. 2k, d, l). In addition, anti-CD147 antibody (clone HIM6) blocked the specificity of CD147-CAR-NK-92MI cell cytotoxicity (Fig. 2m, n). To further validate CD147 as an effective and valid target

for HCC, we examined the cytotoxicity of CD147-CAR-T cells against two different HCC cell lines—HepG2 (Supplementary Fig. 9A) and SK-Hep1 (Supplementary Fig. 9B), using the FFLuc reporter assays.

Expectedly, when cocultured with CD147−/−-SK-Hep1 target cells, the specific lysis of CD147-CAR-T cells had significantly

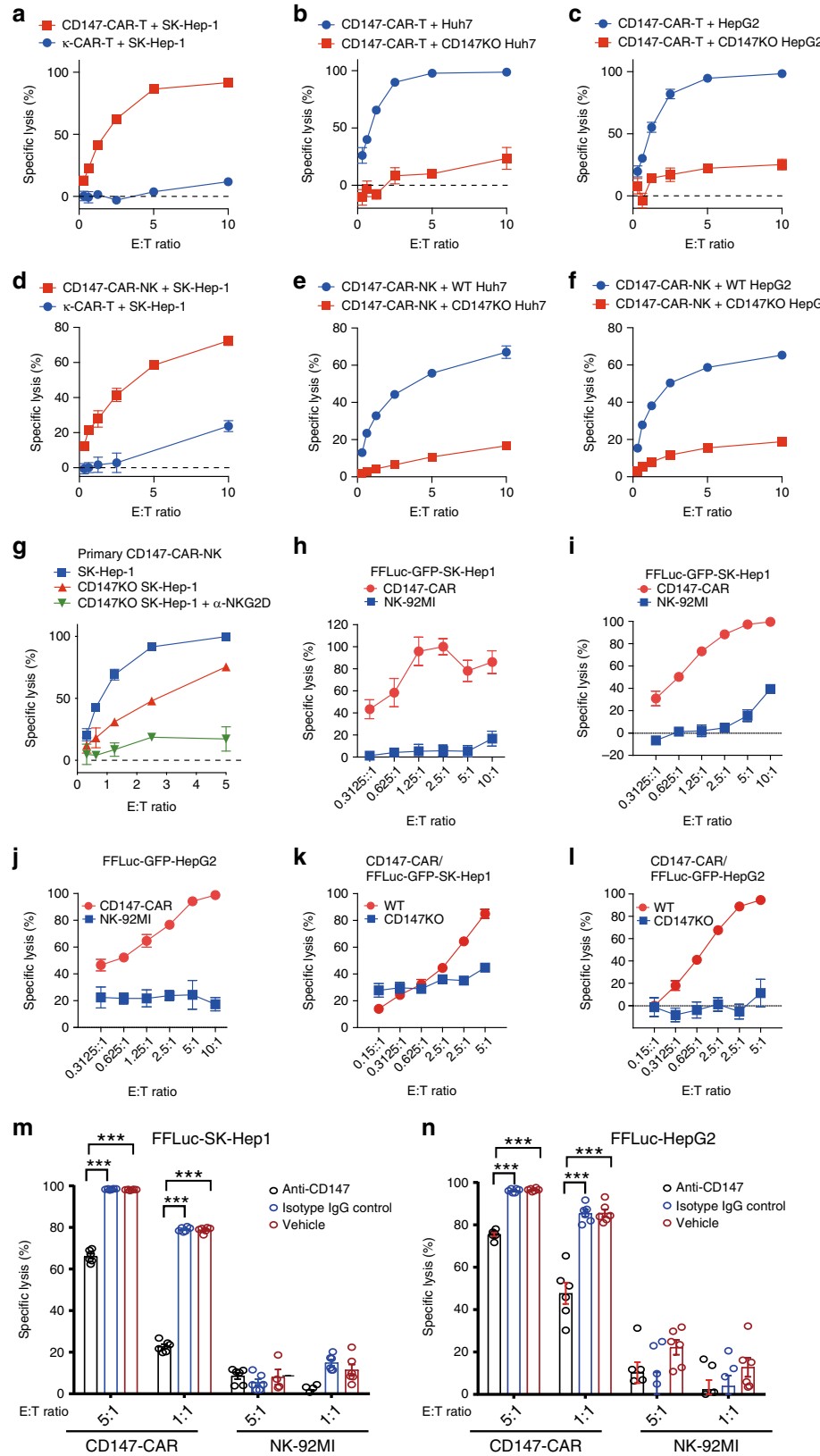

decreased (Supplementary Fig. 9C), which further validated the specific cytotoxicity of CD147-CAR-T cells against CD147+ HCC cell lines. Together, the data indicate that CD147-CAR-modified immune cells, including primary T and NK cells, as well as NK-92MI cells, can kill CD147+ target cells, selectively and specifically.

**CD147-CAR-modified cells control HCC growth in vivo.** To evaluate whether CD147-CAR can kill HCC in vivo, two different xenograft models were used. First, CD147-CAR-modified primary T and NK cells derived from PBMCs were evaluated in an SK-Hep1 xenograft mouse model (Fig. 3). CD147-CAR-modified primary T cells significantly suppress tumor size and prolong

**Fig. 2 CD147-CAR-T and CD147-CAR-NK cells specifically kill CD147[+] tumor cells in vitro. a** Specific cytotoxicity of primary CD147-CAR-T cells was measured by FFLuc reporter assays, compared to κ-CAR-T cells (control group). Data represent the mean ± SEM ($n = 6$). **b, c** Significantly decreased cytotoxicity of CD147-CAR-T cells using knockout-CD147 FFLuc-GFP-SK-Hep1 cell line and HepG2 cell line by FFLuc reporter assays. Data represent the mean ± SEM from three independent experiments. **d** Cytotoxicity of primary CD147-CAR-NK cells was measured by the 4-h standard [51]Cr release assays, compared to κ-CAR-T cells (control group). Data represent the mean ± SEM ($n = 6$). **e, f** Significantly decreased cytotoxicity of primary CD147-CAR-NK cells using knockout-CD147 FFLuc-GFP-SK-Hep1 cell line and HepG2 cell line by the 4-h standard [51]Cr release assays. Data represent the mean ± SEM from three independent experiments. **g** Anti-NKG2D antibody blocks primary CD147-CAR-NK naturally killing to FFLuc-GFP-SK-Hep1. Data represent the mean ± SEM from three independent experiments. **h** Cytotoxicity of CD147-CAR-NK-92MI to the SK-Hep1 was measured by a standard 4-h [51]Cr release assay. Data represent the mean ± SEM ($n = 6$). **i, j** FFLuc reporter system assay for specific killing of FFLuc-GFP-SK-Hep1 and FFLuc-GFP-HepG2 cell lines by CD147-CAR-NK-92MI. Effector cells (CD147-CAR-NK-92MI and NK-92MI) were cocultured with $1 \times 10^4$ FFLuc-GFP-SK-Hep1 (**i**) or FFLuc-GFP-HepG2 (**j**) target cells per well in a 96-well optical-bottom microplate for 6 h. The control groups (blue) used were wild-type NK-92MI incubated with CD147[+] FFLuc-GFP-SK-Hep1 or CD147[+] FFLuc-GFP-HepG2. Data represent the mean ± SEM ($n = 6$). **k, l** Decreased cytotoxicity of CD147-CAR-NK-92MI cells using knockout-CD147 FFLuc-GFP-SK-Hep1 (**k**) and knockout-CD147-FFLuc-GFP-HepG2 (**l**) cell lines by FFLuc reporter system assay. Data represent the mean ± SEM ($n = 3, 6$ for SK-Hep1 and HepG2 groups, respectively). **m, n** Anti-CD147 (clone, HIM6) inhibits the CD147-CAR-NK-92MI-specific lysis effect against and FFLuc-GFP-SK-Hep1 (**m**) and FFLuc-GFP-HepG2 (**n**). Data represent the mean ± SEM from three independent experiments. Two-tailed unpaired Student's $t$ test was employed for all the panels. ***$p < 0.001$; n.s. not significant.

survival (Fig. 3a–d). To further evaluate the efficacy of CD147-CAR-modified primary NK cells, we included a non-transduced (NT) primary NK group as an additional control group (Fig. 3e–h). Expectedly, mice treated with parental NT-NK or phosphate-buffered saline (PBS) vehicle developed rapid disease progression (Fig. 3). In contrast, mice treated with CD147-CAR-modified primary T and NK cells were significantly protected from rapid disease progression and their median survival was prolonged ($P < 0.05$), with comparable body weights among all groups (Fig. 3c–g), indicating the tolerable toxicity of CD147-CAR-modified primary T and NK cells in vivo.

Furthermore, to further develop additional "off-the-shelf" cell therapy strategies, we evaluated the efficacy of CD147-CAR-NK-92MI cells. Due to the malignant nature of NK-92MI, CAR-NK-92MI cells need to be irradiated before being administered to patients[47,63]. We compared the cytotoxicity of non-irradiated and irradiated CD147-CAR-NK-92MI cells by standard 4-h [51]Cr release assays (Supplementary Fig. 10). Comparable cytotoxicity between non-irradiated and irradiated CD147-CAR-NK-92MI cells was observed in vitro (Supplementary Fig. 10). We further compared the efficacies between non-irradiated and irradiated CD147-CAR-NK-92MI cells in the xenograft NOD.Cg-*Prkdc*[scid] *Il2rg*[tm1Wjl]/SzJ (NSG) mouse model (Supplementary Fig. 11). Comparable in vivo efficacies measured by median survival between non-irradiated and irradiated CD147-CAR-NK-92MI-infused mice were observed (Supplementary Fig. 11).

To further evaluate the efficacy of CD147-CAR-NK-92MI cells (injected on days 1, 3, and 5 after tumor implantation) to control tumor growth, disease progression was determined by measuring tumor size (Fig. 4a). Expectedly, mice treated with parental NK-92MI and PBS vehicle control groups developed rapid disease progression (Fig. 4b). In contrast, mice treated with CD147-CAR-NK-92MI cells were significantly protected from rapid disease progression and their median survival was prolonged ($P < 0.01$), with comparable body weights among the different groups (Fig. 4c, d), indicating the tolerable toxicity of CD147-CAR-NK-92MI cells in vivo.

Although cancer cell lines may have significant limitations in their ability to precisely model biology and therapeutic effects[64], patient-derived xenografts[65] (PDXs) models are biologically stable and can mimic human clinic conditions regarding mutational status, gene expression patterns, and tumor heterogeneity. Thus, we employed another xenograft mouse model using metastatic liver cancer tissue from a patient. We tested the ability of CD147-CAR-NK-92MI cells administered on days 0, 4, 8, 11, 15, 22, 25, and 35 after engraftment. The median survival of mice treated with non-irradiated CD147-CAR-NK-92MI cells

was 63 days, which was significantly higher than that of control mice, which was ~42 days. Reduced tumor burden and disease progression were observed in the mice treated with CD147-CAR-NK-92MI cells (Fig. 4e–h), indicating the effectiveness of CD147-CAR-NK-92MI cells in suppressing liver cancer progression in our PDX mouse model.

**HCC-derived CD147-CAR-NK cells kill an CD147[+] HCC cell line.** Due to CD147's broad expression pattern across multiple solid tumor types, CD147 is an attractive target for CD147-CAR-based cancer immunotherapy. In addition to the previous studies[37], we further examined whether CD147 is upregulated in human HCC tissue samples. Different stages of HCC tumor tissue stained strongly positive for CD147, compared to healthy liver tissue (Fig. 5a).

To evaluate whether CD147-CAR-modified primary NK cells directly isolated from HCC-affected livers can kill HCC in vitro, we isolated NK cells from different zones of HCC liver tissue (Fig. 5b), which included a tumor zone, tumor adjacent zone, and a non-tumor zone. Then, we expanded these NK cells (Fig. 5c) and generated CD147-CAR-NK cells using these expanded primary NK cells directly isolated from HCC liver tissues. The transduction efficiency of activated NK cells was generally >70% (Fig. 5d). The anti-tumor activity of CD147-CAR-NK was evaluated against HCC cell lines (Fig. 5e). Together, we conclude that CD147-CAR-redirected primary human liver NK cells kill the CD147[+] target cells, selectively and specifically.

**LogCD147-CAR-T cells kill only CD147[+]GPC3[high] HCC cells.** To mitigate off-tumor toxicity to NT, we assessed how the density of CD147 expression in different types of cells, with a focus on hematopoietic cells, affects the anti-tumor activity of CD147-CAR. We first examined CD147 expression among HepG2, Raji, Daudi, and PBMCs and observed different expression profiles (Supplementary Fig. 12a). Notably, those cells (e.g., PBMCs) expressing low levels of CD147 did not trigger cytotoxicity activity of CD147-CAR-NK-92MI cells even at the high effector and target ratio (E:T ratio) of 10:1 (Supplementary Fig. 12b). These findings suggest that the optimized scFv sequence of anti-CD147 only allows the specific scFv domain to bind with high-expressing CD147, which can mitigate off-tumor toxicity towards NTs that express low levels of CD147.

To further mitigate off-tumor toxicity of CD147-CAR, we used a synNotch receptor that can release transcription factors, which in turn drives expression of a CAR against a different tumor antigen[66]. This "logic-gated" synNotch CAR can only kill dual

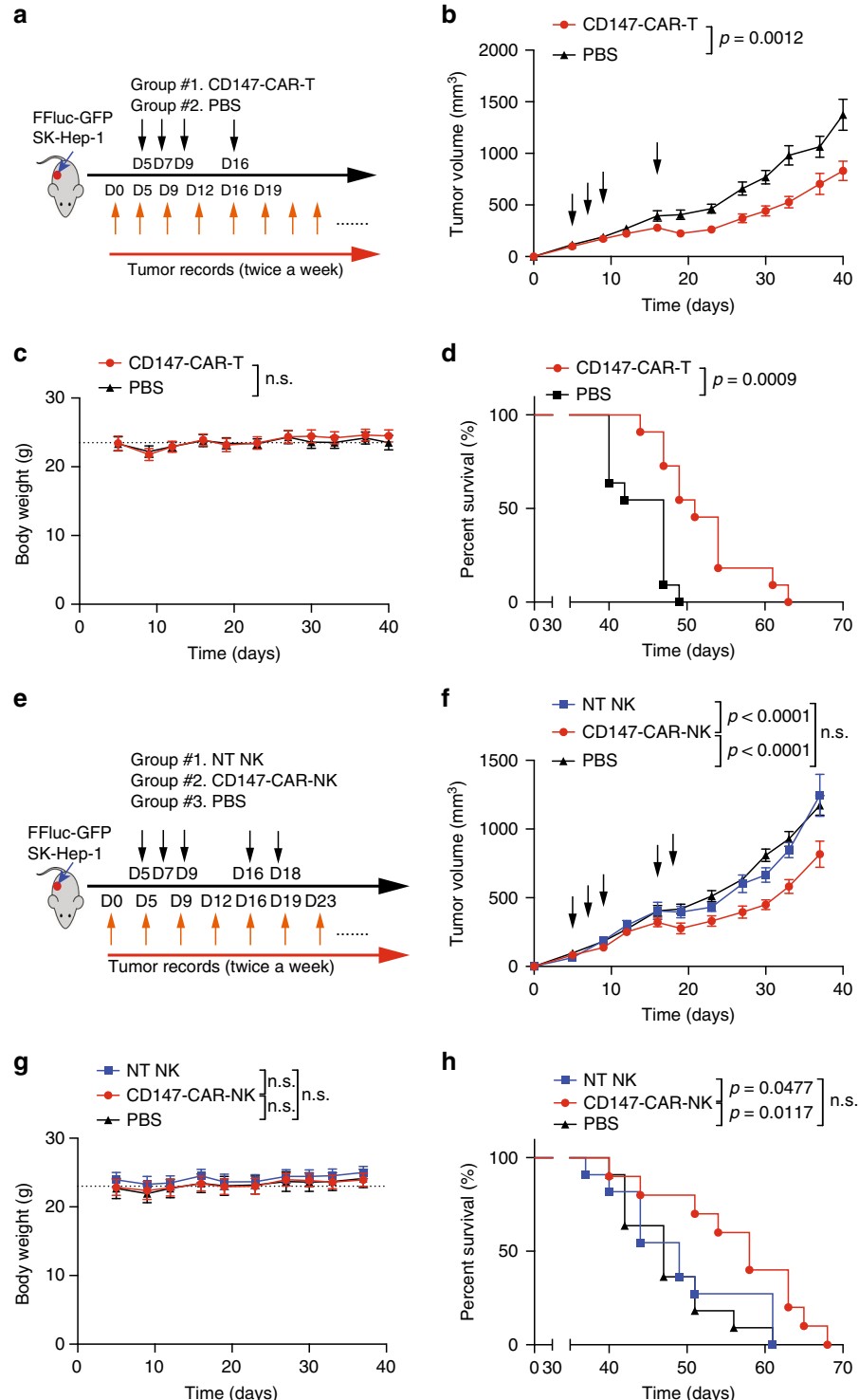

antigen-positive tumor cells, but not single tumor antigen-positive tumor cells (Fig. 6a, b).

Further, we designed a combination approach consisting of GPC3 and CD147 to mitigate off-tumor toxicity. Briefly, we constructed an SFG retroviral vector encoding an anti-GPC3-specific synNotch receptor linking a Gal4VP64 intracellular transcription activation domain. A constitutively expressed enhanced GFP (eGFP) was placed downstream of the GPC3-synNotch to identify transduced cells (Fig. 6a).

A lentiviral vector was constructed in which the anti-CD147-CAR was placed under control of the upstream activating

sequence promoter that can be activated by Gal4VP64 transcription factors released after engagement of the synNotch receptor[66]. A constitutively expressed monomeric read fluorescent protein Cherry (mCherry) was placed downstream of the inducible CD147-CAR to identify transduced cells (Fig. 6a).

Human PBMCs were cotransduced with both lentiviral and retroviral vectors (Fig. 6c). The double-positive cells were verified by eGFP (a marker for anti-GPC3-synNotch) and mCherry (a marker for CD147-CAR) using flow cytometry analysis (Fig. 6d). Four subsets of transduced T cells (including mCherry+-only, GFP+-only, GFP- and mCherry-double-positive, and GFP- and

**Fig. 3 Primary CD147-CAR-T and CD147-CAR-NK cells control progression of HCC in a xenograft mouse model. a** Diagram of experimental design of HCC xenograft model. Briefly, NSG mice were subcutaneous injected with SK-Hep1 cells premixed with equal volume Matrigel (day 0). Tumor burden was determined by size, and once reaching nearly 50 mm$^3$, mice would be randomly grouped on day 4. At day 5 (D5), mice were injected (i.v.) with one dose of $1 \times 10^7$ effector CD147-CAR-T (group #1) cells with IL-2. At days 7, 9, and 16, identical treatments in each group were administered. Day 0 was defined as the initial tumor cell injection time point. **b** Quantification of tumor burden of SK-Hep1 xenografts treated with CD147-CAR-T and PBS, respectively. All results are mean ± SEM. The difference for each group was analyzed by two-way ANOVA analysis. **c** Body weights of each group were measured at the indicated time points. **d** Kaplan–Meier survival curves of tumor-bearing mice after treatment with CD147-CAR-T cells and PBS. The $p$ value was analyzed by log-rank (Mantel–Cox) test. **b–d** $n = 10$ mice per group. **e** Diagram of experimental design for anti-tumor efficacy of primary CD147-CAR-NK in HCC xenograft model. After tumor implantation for 5 days (day 5), the mice were injected (i.v.) with one dose $1 \times 10^7$ effector primary CD147-CAR-NK cells with IL-2. The control groups were injected with the same number of non-transduced primary NK cells with IL-2 or PBS only. At days 5, 7, 9, 16, and 18, identical treatments in each group were administered, as indicated. **f** Quantitative tumor burden of HCC xenograft mice treated with primary CD147-CAR-NK, non-transduced primary NK, and PBS, respectively. All results are mean ± SEM. The difference for each group was analyzed by two-way ANOVA analysis. **g** Body weights of each group were measured at the indicated time points, and analyzed by two-way ANOVA analysis. **h** Kaplan–Meier survival curves of tumor-bearing mice after treatment with primary CD147-CAR-NK, parental primary NK groups, and PBS. The difference for each group was analyzed by log-rank (Mantel–Cox) test. Data represent two separate experiments. The $p$ values are indicated as in comparison of the CD147-CAR-modified cells treated groups with the control groups. **f–h** $n = 11$ mice per group; n.s. not significant.

mCherry-double-negative subsets) were analyzed (Fig. 6e). These transduced T cells were primed by a GPC3$^{high}$CD147$^{-/-}$-HepG2 to induce CD147-CAR expression on the surface (Fig. 6f). Expectedly, we did not observe a leakiness of CAR expression in transduced primary T cells in the absence of synNotch signaling. This observation was further verified by a gamma secretase inhibitor (MK-0752, Notch signaling inhibitor) treatment assay (Supplementary Fig. 13). The CD147-CAR expression on GPC3-synNotch-GFP- and CD147-CAR-mCherry-double-positive T cells was dramatically inhibited upon MK-0752 treatment (Supplementary Fig. 13).

Following GPC3-synNotch-GFP and CD147-CAR-mCherry cotransduction of T cells and primed by the CD147$^{ko}$GPC3$^{high}$-HepG2, the cytotoxic activities of transduced T cells were triggered by different subsets of HCC cell lines for 2 h to assess specific tumor-killing efficacy. The different subsets of HCC cell lines were: CD147$^+$GPC3$^{high}$ HepG2, CD147$^{ko}$GPC3$^{high}$ HepG2, CD147$^+$GPC3$^{low}$ HepG2, and CD147$^{ko}$GPC3$^{low}$ HepG2 (Fig. 6g). Phorbol-12-myristate-13-acetate/ionomycin was used as a positive control.

GPC3-synNotch-GFP- and CD147-CAR-mCherry-double-positive T cells that were primed with CD147$^{ko}$GPC3$^{high}$ HepG2 cells can be specifically activated by the CD147$^+$GPC3$^{high}$ HepG2 cells (Fig. 6h), which was quantified by CD107a surface expression when cocultured with different target cell lines. Similar results were obtained when we cotransduced a myc-tagged CD147-specific-synNotch-GFP and inducible GPC3-CAR-mCherry into T cells (Supplementary Fig. 14).

Together, the data suggest that GPC3-synNotch-inducible CD147-CAR-T cells can specifically be activated by only CD147$^+$GPC3$^{high}$ HepG2 cells, but not by CD147$^{ko}$GPC3$^+$ or CD147$^+$GPC3$^{low}$ HepG2 cells. These activated CD147-CAR-T cells can kill CD147$^+$GPC3$^{high}$ HepG2 cells, but not CD147$^{ko}$GPC3$^+$ HepG2 cells (Supplementary Fig. 15).

To further test whether the GPC3-synNotch-inducible CD147-CAR cells can control progression of HCC in vivo, $1.3 \times 10^6$ irradiated GPC3-synNotch-inducible CD147-CAR-NK-92MI cells with $2 \times 10^4$ IU IL-2 were adoptively transferred intraperitoneally (i.p.) into the CD147$^+$GPC3$^{high}$ HepG2 and CD147$^+$GPC3$^{low}$ HepG2 inoculated xenograft NSG mice (Supplementary Fig. 16). The CD147$^+$GPC3$^{low}$ HepG2 cells cannot effectively prime the GPC3-synNotch-inducible CD147-CAR-NK-92MI cells due to the low surface expression of GPC3 molecules. As expected, tumor burden from the CD147$^+$GPC3$^{high}$ HepG2 inoculated mice treated with GPC3-synNotch-inducible CD147-CAR-NK-92MI cells was reduced, compared with that of mice bearing CD147$^+$GPC3$^{low}$ HepG2 tumor cells (Supplementary

Fig. 16). These data reveal GPC3-synNotch-inducible CD147-CAR-NK-92MI cells are potently cytotoxic and can control progression of HCC in vivo.

Previous studies show that epithelial and fetal tissues under normal conditions express low levels of CD147 molecule[34]. CD147 is expressed on a variety of cell types (e.g., hematopoietic, epithelial, and endothelial cells) and at varying levels[33]. To test whether the toxicity that may result from GPC3-synNotch-inducible CD147-CAR targeting of CD147 in healthy tissues, we generated a hCD147TG mouse model to test the on-target/off-tumor toxicity of GPC3-synNotch-inducible CD147-CAR-NK-92MI treatment of HCC (Supplementary Fig. 17). Genotyping confirmed the successful generation of hCD147TG (Supplementary Fig. 17A). IHC showed human CD147 (hCD147) expression on liver tissues on hCD147TG mice (Supplementary Fig. 17B). The genotyping products were verified by sequencing. To test whether the on-target/off-tumor toxicity of GPC3-synNotch-inducible CD147-CAR-NK-92MI cells, non-tumor-bearing hCD147TG mice were treated with in vitro activated, primed, and irradiated GPC3-synNotch-inducible CD147-CAR-NK-92MI cells. The body weight of treated animal was recorded, as well as appearance and activity. Notably, the body weight of treated animal slightly increased after three doses of activated GPC3-synNotch-inducible CD147-CAR-NK-92MI cell injection (Supplementary Fig. S18). Thus, we conclude that these activated GPC3-synNotch-inducible CD147-CAR-NK-92MI cells do not cause severe on-target/off-tumor toxicity in vivo.

## Discussion

We show that human primary T cells, primary NK cells, and NK-92MI cell line transduced with CD147-CAR molecules can specifically kill malignant HCC cell lines in vitro, and control progression of HCC in a xenograft mouse model, a PDX mouse model, and a hCD147TG mouse model. We also observed transient fratricide during CD147-CAR-NK-92MI expansion due to the presence of a low CD147 expressing subset of parental NK-92MI. However, the transient fratricide did not affect the cytotoxicity, expansion, and proliferation of CD147-CAR-NK-92MI cells. Instead, this transient fratricide selected for the CD147$^-$ or relatively lower expressing subset of CD147-CAR-NK-92MI and facilitated its expansion. The synNotch receptor logic-gated GPC3 and CD147-CAR can further mitigate on-target/off-tumor toxicity to NT in a hCD147TG mouse model. These findings support the clinical development of CD147-CAR for HCC immunotherapy.

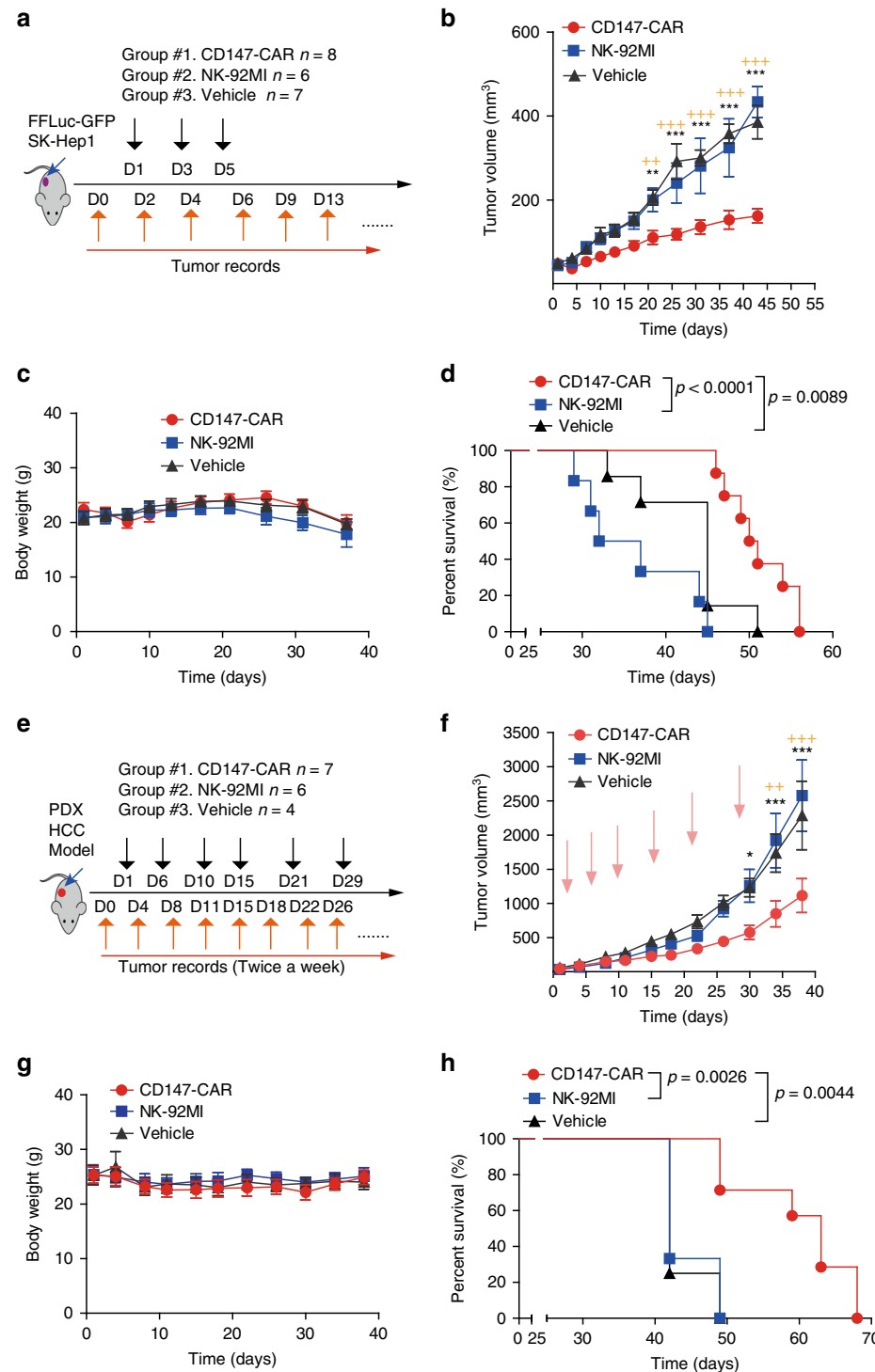

Recently, an agent against CD147, called ABX-CBL (gavilimomab, a hybridoma-generated murine IgM monoclonal antibody), has been proposed to specifically inhibit graft rejection after CD147 blockade was tested for the treatment of steroid-resistant acute GVHD[67].

Metuzumab (a recombinant, glycoengineered human/mouse chimeric IgG1 monoclonal antibody against the extracellular C2 domain of hCD147, HAb18G) was developed to optimize antibody-dependent cellular cytotoxicity[68]. A $^{131}$I-labeled HAb18 F(ab′)2 (Licartin) targeting CD147 has been reported as an efficacious and safe treatment option for patients with primary HCC[45].

However, current antibody-based immunotherapy strategies lack a sustained effect, given the short half-life of IgM antibody (~5–6 days) and of IgG1 (~21 days). The second disadvantage of using an antibody is the development of human anti-mouse antibodies, which limits the application of murine antibodies in clinical contexts.

CD147-CAR-T cells or NK cells have better longevity and the additional benefit of self-replication. Some CAR-T cells exhibit long-lived, memory-like phenotypes and reside in the bone marrow for more than several years[69–71,72]. Manageable toxicities of CAR-modified cells also contribute to the durable anti-tumor responses.

**Fig. 4 Non-irradiated CD147-CAR-NK-92MI cells control progression of HCC in a xenograft mouse model and a PDX mouse model. a** Diagram of experimental design of HCC xenograft model. At day 0, tumor burden (~50 mm³) was determined and mice were randomly grouped. Of note, day 0 was defined as the day before CAR treatment. **b** Quantification of tumor burden of SK-Hep1 xenografts treated with CD147-CAR-NK-92MI, parental NK-92MI cells, and PBS, respectively. All results are mean ± SEM. The difference for each group ($n = 6–8$ per group) was analyzed by two-way ANOVA analysis. *$P < 0.05$, **$p < 0.01$, and *** $p < 0.001$ are indicated as in comparison of the CD147-CAR-treated group with the NK-92MI-treated group. $^+P < 0.05$, $^{++}p < 0.01$, and $^{+++}p < 0.001$ are indicated in comparison of the CD147-CAR-treated group with the vehicle control-treated group. **c** Body weights of each group were measured at the indicated time points. **d** Kaplan–Meier survival curves of tumor-bearing mice after treatment with CD147-CAR-NK-92MI cells ($n = 8$), parental NK-92MI group ($n = 6$), and PBS ($n = 7$). **e** Diagram of experimental design for anti-tumor efficacy of CD147-CAR-NK-92MI in a liver PDX model. After tumor implantation for 4 weeks (day 1), tumor burden was determined (~50 mm³) and mice were randomly grouped. **f** Quantitative tumor burden of PDX mice treated with CD147-CAR-NK-92MI cells, parental NK-92MI cells (control group), and PBS (vehicle control group), respectively. All results are mean ± SEM. The difference for each group was analyzed by two-way ANOVA analysis. *$P < 0.05$, **$p < 0.01$, and ***$p < 0.001$ are indicated as in comparison of the CD147-CAR-treated group with the NK-92MI-treated group. The $^{++}p < 0.01$, and $^{+++}p < 0.001$ are indicated in comparison of the CD147-CAR-treated group with the vehicle control-treated group. **g** Body weights of each group were measured at the indicated time points. **h** Kaplan–Meier survival curves of tumor-bearing mice after treatment with CD147-CAR-NK-92MI cells ($n = 7$), parental NK-92MI groups ($n = 6$), and PBS ($n = 4$). The $p$ value analysis by log-rank (Mantel–Cox) test. Data are from two experiments.

Successful manufacture and time-sensitive infusion of autologous CAR-modified immune cells is the key to an effective CAR-T or CAR-NK cell-based immunotherapy[47]. However, it is challenging to produce CAR-T or CAR-NK cells from patients with high-risk solid tumors[73]. Therefore, the CD147-CAR-NK cells developed here may provide an allogeneic off-the-shelf cell-based product. A pressing issue existing in the field of immunotherapy is uncertainty around whether an off-the-shelf universal CAR product can be developed[47]. CAR-T cell products from individuals are costly and take a considerable amount of time to prepare. Generation of off-the-shelf CAR-T or NK cell products will significantly reduce the cost of immunotherapy. Using clustered regularly interspaced short palindromic repeats (CRISPR) and the CRISPR-associated protein 9 (Cas9) technique[74], as well as other gene-editing technologies to knockout endogenous TCR's and HLA class I molecules for universal CAR-modified T cell generation are still in preclinical phases[47,75]. Unfortunately, most of these strategies are still in the early phase.

Compared to other CAR-T cell immunotherapy for HCC, the proposed CD147-CAR-modified immune cells possess three distinct advantages. First, we identified that the CD147 molecule can be used as an effective and valid target for CAR-mediated immunotherapy in vitro and in vivo. Second, CD147-CAR-NK-92MI is feasible with high specificity and tolerable toxicity. Third, CD147-CAR-NK cells have a potential for off-the-shelf products for various tumors with CD147 upregulation.

Since CD147 is also expressed on several organs with varying expression levels, there is a concern that infused CD147-CAR-modified immune cells may also target other organs. Therefore, such on-target/off-tumor toxicity should be tested in future clinical trials. Our data show that GPC3-synNotch-inducible CD147-CAR-T cells or CD147-synNotch-inducible GPC3-CAR-T or GPC3-CAR-NK cells can be specifically activated by CD147⁺GPC3^high HepG2 cells, but not CD147^koGPC3⁺ or CD147⁺GPC3^low HepG2 cells. Meanwhile, this on-target/off-tumor toxicity of CD147-CAR-NK or CD147-CAR-T cell therapy may be manageable by introducing suicide genes[76], using messenger RNA transfection[77], a split, universal, and programmable CAR system[78], intratumoral injections of CAR-modified cells[79], or a synNotch CAR design[80–82].

In conclusion, we demonstrate that CD147-CAR can effectively redirect human immune cells (primary T, primary NK, and NK-92MI cells) to target malignant HCC cells with limited fratricide in vitro and in vivo. Meanwhile, the hCD147TG mouse model developed in this study will be useful in testing hCD147-CAR cell safety. This approach may provide treatment options for patients with HCC. Importantly, future clinical trials using CD147-CAR,

as well as transgenic animal models, will further evaluate the on-target/off-tumor toxicity.

## Methods

**Antibodies and reagents.** Purified anti-CD247 (also known as T cell surface glycoprotein CD3ζ) antibody (clone 6B10.2, BioLegend), purified anti-hCD147, fluorescein isothiocyanate (FITC)-conjugated anti-hCD147 (clone HIM6, BioLegend), phycoerythrin (PE)- or allophycocyanin (APC)-conjugated anti-human CD3 antibody (clone OKT3, BioLegend), FITC- or BV510-conjugated anti-human CD56 antibody (clone HCD56, BioLegend), PE-conjugated anti-human CD69 antibody (clone FN50, BioLegend), APC/Fire 750-conjugated anti-human CD226 antibody (also known as DNAM-1, clone 11A8, BioLegend), APC/Fire 750-conjugated anti-human KLRG1 (MAFA) antibody (clone SA231A2, BioLegend), BV421-conjugated anti-human CD335 (NKp46) antibody (clone 9E2, BioLegend), PE/Cy7-conjugated anti-human CD158b (KIR2DL2/L3, BioLegend) antibody (clone DX27, BioLegend), PE/Cy7-conjugated anti-human CD244 (2B4) antibody (clone C1.7, BioLegend), PE-conjugated anti-human CD152 (CTLA-4) antibody (clone BNI3), APC-conjugated anti-human CD366 (Tim-3) antibody (clone F38-2E2), PerCP/Cy5.5 anti-human TIGIT (VSTM3) antibody (clone A15153G), FITC-conjugated anti-human CD223 (LAG-3) antibody (clone 11C3C65, BioLegend), and PerCP/Cy5.5-conjugated anti-human CD94 (clone DX22, BioLegend) were purchased from BioLegend (San Diego, CA, USA).

APC-conjugated anti-human CD16 antibody (clone B73.1, BD Biosciences), FITC-conjugated anti-human CD3 antibody (clone UCHT1, BD Biosciences), BV480-conjugated anti-human CD85j antibody (LIR-1) antibody (clone GHI/75, BD Biosciences), BV711-conjugated anti-human CD314 (NKG2D) antibody (clone 1D11, BD Biosciences), and PE- or FITC-conjugated anti-human CD107a antibody (clone H4A3, BD Biosciences) were purchased from BD Biosciences (San Jose, CA, USA).

FITC-conjugated anti-human KIR/CD158 antibody (clone 180704, R&D Systems), PE-conjugated anti-human KIR2DL1/KIR2DS5 antibody (clone 143211, R&D Systems), APC-conjugated anti-human KIR3DL1 antibody (clone DX9, R&D Systems), AF405-conjugated anti-human KIR3DL2/CD158k antibody (clone 539304, R&D Systems), APC-conjugated anti-human NKG2A/CD159a antibody (clone 131411, R&D Systems), and PE-conjugated anti-human NKG2C/CD159c antibody (clone 134591, R&D Systems) were purchased from R&D Systems. AF647 Goat anti-human IgG F(ab′)² fragment antibody was purchased from Jackson ImmunoResearch (West Grove, PA, USA). All primary antibodies used for flow cytometry were used at the same final concentration (1:100 dilution), unless otherwise specified. For immunofluorescence assays, the concentrations of primary and secondary antibodies were 1:100 and 1:1000 dilutions, respectively.

**Bioinformatic analysis from public cancer patient database.** Patient survival data and RSEM (RNA-sequencing by expectation maximization) normalized expression datasets about CD147 were generated from TCGA and were downloaded from OncoLnc (http://www.oncolnc.org/). Data were plotted for Kaplan–Meier curves using GraphPad Prism 5.0 (GraphPad). RSEM normalized expression datasets derived from TCGA come from FireHose Broad GDAC, which was developed by The Broad Institute (https://gdac.broadinstitute.org/). Figures also were generated by GraphPad Prism 5.0.

**Cell lines.** NK-92, NK-92MI, 293T, K562, Daudi cell, SK-Hep1, and HepG2 cell lines were purchased from American Type Culture Collection (ATCC). T2 cell line was a gift from Dr. June Kan-Mitchell (University of Texas at El Paso, El Paso, TX). MDA-MB-231 was a gift from Dr. Mien-Chie Hung (University of Texas MD Anderson Cancer Center, Houston, TX). The 721.221 cell line was a gift from Dr. Eric O. Long (National Institute of Allergy and Infectious Diseases, NIH). To

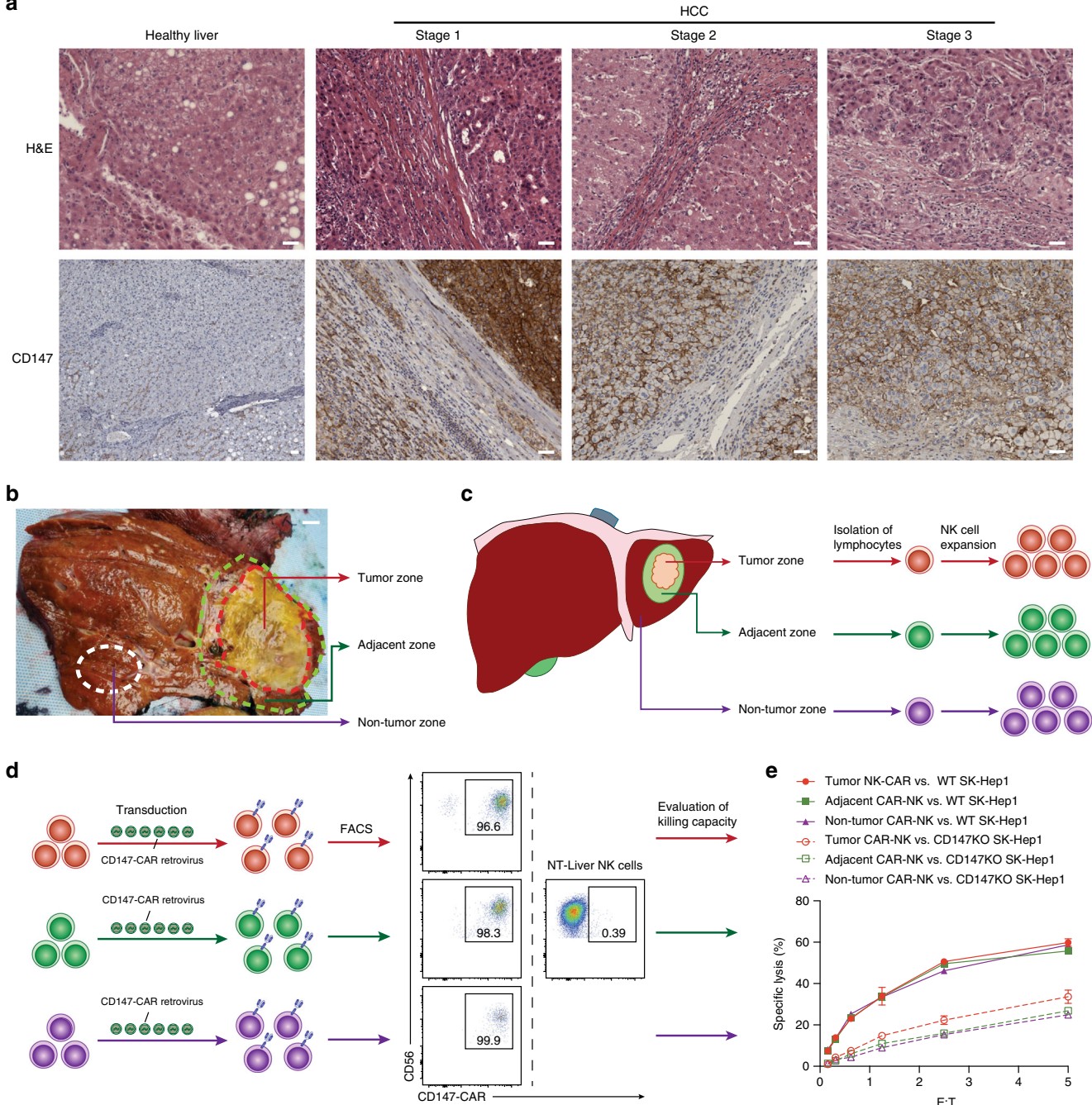

**Fig. 5 Patient-derived Primary CD147-CAR-NK cells specifically kill CD147$^+$ tumor cells in vitro. a, b** Representative H&E and IHC staining of liver samples from different stages of one HCC patient. Scale bar, 200 μm. **c** Diagram of experimental design of HCC sample acquisition from different areas of liver cancer tissues. Briefly, three regions of interest (tumor zone, adjacent zone, and non-tumor zone) were obtained. Primary NK cells were isolated from these zones, indicated by different colors. Scale bar, 2 cm. **d** Flow cytometry analysis of CD147-CAR$^+$ primary NK cells from different zones of liver tissues. **e** Cytotoxicity of primary CD147-CAR-NK cells was measured by 4-h standard Cr$^{51}$ release assays. All results are mean ± SEM. Data are from at least two experiments.

establish the FFluc-GFP$^+$, CD147$^+$ HepG2, and SK-Hep1 cells were transduced with a lentiviral vector encoding FFLuc-GFP. The GFP$^+$ cells were sorted by BD FACS Aria III. The protocol for collection of peripheral blood from healthy donors and liver tissue from HCC patients were approved by the institutional review board (IRB) and ethics review committees at the Rutgers-New Jersey Medical School (Newark, NJ). Where necessary, patient consent was obtained before obtaining clinical samples.

**HCC tumor and liver-derived CAR-NK cell culture and generation.** A patient scheduled for an orthotopic liver transplant at the University Hospital was selected as a suitable candidate for the purposes of generating primary liver CAR-NK cells under the approved IRB protocols. The recipient's liver tissues were analyzed at the department of pathology, immunology, and laboratory medicine in Rutgers-New Jersey Medical School. Sample tissues from the grossed tumor, adjacent, and distant unaffected areas were harvested. After briefly mincing the tissue using sterile forceps, the tissue was further dissociated using a GentleMACS Octo with heater (Miltenyi Biotec) for 45 min in collagenase IV (Thermo Fisher) using a custom dissociation protocol. Next, the resulting suspension was triturated through a 70 μm cell strainer (Corning) before washing with 30% Percoll (GE Healthcare) and spinning at $400 \times g$ for 5 min at room temperature. The supernatant was aspirated, and the cell pellet was resuspended in complete RPMI-1640 and carefully layered

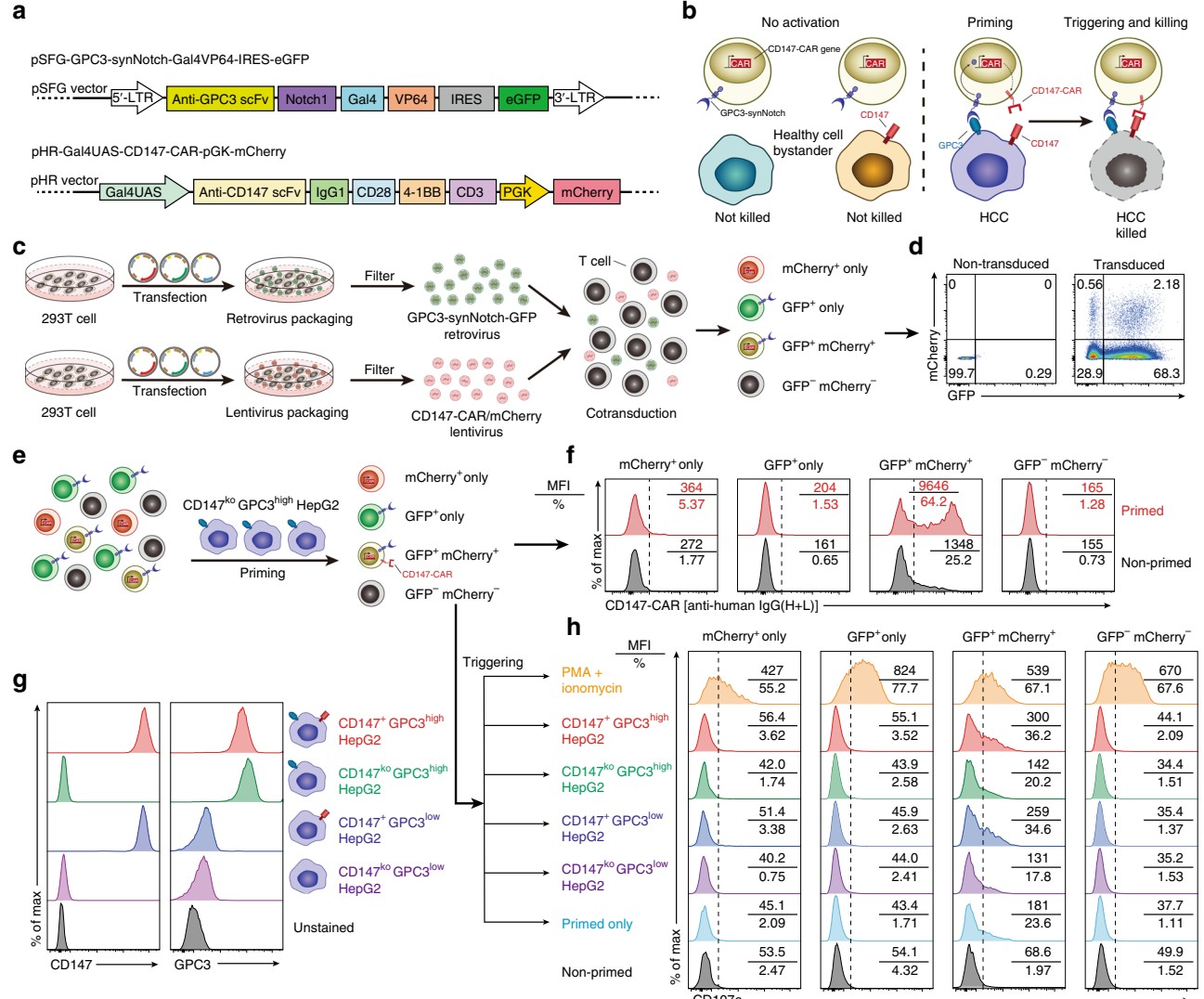

**Fig. 6 GPC3-synNotch-inducible CD147-CAR-T cells selectively target GPC3$^+$CD147$^+$-double-positive, but not single-positive HepG2 cells.**
**a** Schematic design of GPC3-Gal4VP64-synNotch receptor in SFG retroviral vector and CD147-CAR based on the pHR lentiviral vector. The SFG retroviral vector contains eGFP, which can be used as a marker for selecting GPC3-Gal4VP64-synNotch$^+$ cells. The pHR lentiviral vector contains mCherry, which can be used as a marker for selecting CD147-CAR$^+$ cells. **b** Schematic design of "logic-gated" GPC3-synNotch and CD147-CAR showing induced cytotoxicity when both antigens are coexpressed, but not when they are separately expressed on bystander or healthy cells. **c, d** Schematic experimental design of GPC3-synNotch-GFP and CD147-CAR-mCherry vectors cotransduced T cells (**c**) and representative flow cytometric analysis of GPC3-synNotch-GFP and CD147-CAR-mCherry expression (**d**). **e** Schematic experimental design of GPC3-synNotch-GFP and CD147-CAR-mCherry vectors cotransduced into T cells, priming by GPC3$^{high}$CD147$^{low}$ HepG2 cell line, and followed by CD147-CAR expression analysis among different subsets of transduced T cells, including mCherry$^+$-only, GFP$^+$-only, GFP- and mCherry-double-positive, and GFP- and mCherry-double-negative subsets. **f** Representative flow cytometric analysis of CD147-CAR expression on the surface of different subsets of transduced T cells. Both mean fluorescence intensity (MFI) and percentage of CD147-CAR are displayed in each representative flow cytometric chart. **g** Representative flow cytometric analysis of CD147 and GPC3 expression on HepG2 tumor cell lines. **h** Quantitative analysis of surface CD107a expression on different subsets of transduced T cells after "primed and triggered" protocol by different HepG2 tumor cell lines for 2 h. Data are representative of two independent experiments.

over Lymphocyte Isolation medium (Corning) for density gradient centrifugation at $400 \times g$ for 23 min without acceleration or brakes. The Ficoll layer was carefully collected and washed with complete medium before putting into culture to expand liver-resident NK cells. The protocol for collection of liver tissue from HCC patients was approved by the Rutgers IRB and ethics review committees at Rutgers-New Jersey Medical School (Newark, NJ). Where necessary, patient consent was obtained before obtaining clinical samples. NK and CAR-NK cells were expanded with 100-Gy irradiated 721.221-mIL21 cells and supplemented with 200 U/mL IL-2 and 5 ng/mL IL-15 (Peprotech, Rocky Hill, CT, USA)[83].

**NK-92MI cell culture and generation of CAR-modified NK-92MI cells.** NK-92MI cell line was purchased from ATCC (CRL-2408, USA). NK-92MI, an IL-2-independent NK cell line, is derived from NK-92 (ATCC CRL-2407) cell line[49] stably expressed with human IL-2 cDNA[59,62]. NK-92MI cells were maintained in

the specific NK-92MI culture medium (α-minimum essential medium) without ribonucleosides or deoxyribonucleosides, but with 2 mM L-glutamine and 1.5 g/L sodium bicarbonate. To make the complete growth medium, the following components were added to the base medium: 0.2 mM inositol, 0.1 mM 2-mercaptoethanol, 0.02 mM folic acid, horse serum to a final concentration of 12.5%, and fetal bovine serum (FBS) to a final concentration of 12.5%. NK-92MI cells were transduced with retroviral supernatants on day 3 in plates coated with recombinant fibronectin fragment (RetroNectin; Takara, Japan). After transduction, NK cells were maintained using minimal concentration IL-2 (40 ng/mL). To check the percentage of CD147-CAR expression on NK-92MI cells, these cells were stained for CD3 and CD56 to stain NK cells, followed by flow cytometry analysis.

**Generation of CD147$^{-/-}$ cell line.** To generate the CD147$^{-/-}$ HCC cell lines, we employed a lentiviral delivery system using guide RNA targeting the

CD147 sequence: #1 (5-TTGACATCGTTGGCCACCGC-3) and #3 (5-GTGGACGCAGATGACCGCTC-3). Lentivirus was produced in HEK 293T by transfecting Lenti-CRISPR v2 with packaging plasmids pSPAX2 and pMD2G. After 3 days, supernatants were filtered (0.45 μm) and incubated with HCC cells and 8 μg/mL polybrene (Sigma). After 48 h of incubation, transduced cells were changed to a fresh medium and selected with 8.0 μg/mL puromycin for 5 days. WB and flow cytometry analysis were performed to confirm the phenotype of the knockout cell lines.

**Plasmid construction and retrovirus production.** A codon-optimized DNA fragment was synthesized by GENEWIZ encoding the CD147-specific scFv from the murine 5F6 clone and subcloned into the SFG retroviral vector retroviral backbone in-frame with the hinge component of human IgG1, CD28 transmembrane domain, intracellular domain CD28 plus 4-1BB, and the ζ chain of the human TCR/CD3 complex. To produce CD147-CAR retrovirus, 293T cells were transfected with a combination of plasmids containing CD147-CAR, RDF, and PegPam3, as previously described[60]. The construct of CD19-CD28-CAR and CD19-4-1BB-CAR has been previously described[61].

To generate the anti-GPC3-synNotch-induced receptor vector, anti-GPC3 (mouse GC33 clone) scFv that can specifically bind with human GPC3 antigen was synthesized by GENEWIZ. The sequence encoding a signal peptide and a myc-tag at the N-terminal were fused with the synNotch-Gal4VP64-induced element derived from (Addgene plasmid #79125) by overlap PCR. The fragments were inserted into the SFG gamma retrovirus vector, which were digested by restriction endonucleases NcoI and XhoI.

For construction of the anti-CD147-CAR-mCherry vector, the entire CD147-CAR element was inserted into pHR_Gal4UAS_pGK_mCherry (Addgene plasmids #79124), which was digested by restriction endonucleases MluI and NdeI. The expression of the mCherry gene was under control of the pGK promoter, as described previously[66]. In this strategy, eGFP- and mCherry-double-positive cells were gated as synNotch CAR-modified cells for further analysis and functional evaluation.

To generate the anti-CD147-synNotch-induced receptor vector, anti-CD147 scFv was fused with the synNotch-Gal4VP64-induced element derived from pHR_PGK_antiCD19_synNotch_Gal4VP64 (Addgene plasmid #79125) by overlap PCR. A myc-tag was added to the N-terminal. The fragments were inserted into the SFG gamma retrovirus vector after the signal peptide, which were digested by restriction endonucleases SalI and MluI.

For construction of the anti-GPC3-CAR-mCherry vector, the entire GPC3-CAR element, which contains anti-GPC3 (mouse GC33 clone) scFv, was inserted into pHR_Gal4UAS_pGK_mCherry (Addgene plasmid #79124), which was digested by restriction endonucleases MluI and NdeI. Expression of the mCherry gene was under control of the pGK promoter. In this strategy, myc-tagged and mCherry-double-positive cells were gated as synNotch CAR-modified cells for further analysis and functional evaluation. All constructs and primers are summarized in Supplementary Table 1.

**Transduction of primary NK cells with CD147-CAR.** NK cells were harvested on day 7 of expansion and transduced with CD147-CAR retrovirus in plates coated with RetroNectin. Two days later, cells were transferred to G-Rex 6 multi-well cell culture plates and maintained in 35 mL of complete NK culture media according to ATCC with 200 U/mL IL-2 (PeproTech). The medium was changed every 3–4 days and $2 \times 10^7$ cells were kept in each well for continued culture at each time. Total cell numbers were counted using trypan blue exclusion. To check the percentage of NK cells and the expression of CAR, cells were stained for CD3, CD56, and IgG F (ab′)$_2$, and analyzed by flow cytometry.

**Flow cytometry.** CAR-modified immune cells were stained with fluorescence-conjugated antibodies in FACS staining buffer (PBS with 1% FBS) on ice for 30 min on ice in the dark. Then, cells were washed with PBS, and analyzed on either a FACS LSRII or an LSR Fortessa flow cytometer (BD). Photomultiplier tube voltages were adjusted using the FACS Diva (BD) software and compensation values were calculated before data collection. Flow data were then acquired using FACS Diva (BD) and analyzed using FlowJo (Tree Star).

For flow cytometry single live cell sorting, sample cells were stained with fluorescence-conjugated antibodies with (RPMI-1640 with 1% FBS) on ice for 30 min, washed with PBS twice, resuspended in completed culture medium, and sorted by SORP BD FACS Aria III. After sorting, collection samples were washed with prewarmed medium once, and cultured for use.

**CAR-NK degranulation (CD107a) assay.** CD107a degranulation assay was described previously[84,85]. Briefly, CAR-NK cells ($1 \times 10^5$) were incubated with target cells in U-bottomed 96-well plates in complete NK-92MI culture media at 37 °C for 4 h, 10 h, or overnight in the presence of CD107a with GolgiStop (BD). Then, cells were stained for CD3 and CD56 for 30 min on ice. The expression of CD107a was analyzed by flow cytometry.

**Cytokine release assays.** The IFNγ and TNF-α cytokines secreted by CAR-NK cells were measured by a commercial ELISA kit (Invitrogen-Thermo Fisher Scientific) as per the manufacturer's protocol.

**$^{51}$Cr release assay.** To evaluate the cytotoxic activity of CAR-NK cell, the standard 4-h $^{51}$Cr release assay was used. Briefly, target cells were labeled with $^{51}$Cr at 37 °C for 2 h and then resuspended at $2 \times 10^5$/mL in NK-92MI culture medium with 10% FBS without IL-2. Then, $2 \times 10^4$ target cells were incubated with serially diluted CAR-NK cells at 37 °C for 4 h. After centrifugation, the supernatants were collected and the released $^{51}$Cr was measured with a gamma counter (Wallac, Turku, Finland). The cytotoxicity (as a percentage) was calculated as follows: [(sample − spontaneous release)/(maximum release − spontaneous release)] × 100.

**FFLuc reporter assay.** To quantify the cytotoxicity of CAR-modified immune cells, we also developed the FFLuc reporter system assay. Briefly, on day 1, target cells were preseeded at 2 or $3 \times 10^4$ target cells/well (FFluc-GFP stably transduced cell) onto an optical 96-well plate (Greiner Bio-One™ No.: 655098) in 100 μL/well of the target cell's full nutrition medium and incubated at 37 °C with 5% $CO_2$, overnight. The next day, serial dilutions of effector cell were prepared according to the effector to target ratio and added into each well (100 μL/well). The reaction was incubated at 37 °C with 5% $CO_2$ for 4 h and then the supernatant was gently discarded. One hundred microliters of working concentration D-luciferin was added to each well and incubated at 37 °C with 5% $CO_2$ for 5 min, with the lights turned off. A microplate reader (PerkinElmer, USA) was used to quantify the data. The data were quantified by converting the obtained values to percentage of specific lysis by the following equation: specific lysis percentage (%) = $[1 - (S - E)/(T - M)] \times 100$, where $S$ is the value of luminescence of the sample well, $E$ is the value of luminescence of the effector cell well compared to the sample well, $T$ is the mean value of luminescence of target cell wells, and $M$ is the mean value of luminescence of blank medium wells.

**Animal studies.** All animal experiments have been approved by the Rutgers Institutional Animal Care and Use Committee (IACUC, PROTO201800200). NSG mice from The Jackson Laboratory (Bar Harbor, ME) were used for all, in vivo, experiments. To establish an HCC cell line xenograft model, both male and female NSG mice (8-week-old) were injected subcutaneously with $4 \times 10^6$ SK-Hep1 cells in 100 μL of PBS Corning® Matrigel® Matrix in the right flank. When the tumor burden reached 40–50 mm$^3$, mice were randomly allocated into three groups. Beginning treatment on day 1, the mice were injected (intravenously (i.v.)) with $5 \times 10^6$ CD147-CAR-NK-92MI cells in 100 μL of PBS. Control groups were infused with either parental NK-92MI or vehicle (PBS). The next day (day 2), all animals were injected (i.v.) with IL-2 (20,000 U/mouse). Animal weight and tumor burden were collected, twice a week. The tumor size was measured by a caliper and the greatest longitudinal diameter (length) and the greatest transverse diameter (width) were recorded. Tumor sizes based on caliper measurements were calculated by the modified ellipsoidal formula. The tumor size was calculated as follows: tumor size (mm$^3$) = ½ (length × width$^2$). When the tumor burden was >2000 mm$^3$ or the animal's weight reduced >20%, mice were euthanized according to Rutgers IACUC guidelines. The animal survival data were recorded simultaneously.

For the PDX model, patient metastatic liver cancer animal models were developed and obtained by The Jackson Laboratory. Briefly, fresh PDX specimens were implanted subcutaneously into the flanks of 6–8-week-old NSG mice. After the tumor burden reached 40–50 mm$^3$, mice were randomly allocated into three groups for further analysis. The treatment procedures used in this study had been described in the Fig. 4e. Xenografts specimens were fixed with 10% formalin, embedded in paraffin for cutting, and processed for IHC staining or were directly frozen into liquid nitrogen for further analysis.

**Generation of hCD147TG transgenic mice.** We developed a humanized NSG mouse model in which hCD147-specific CAR-T/NK cells could be adoptively transferred into mice whose normal cells and tissue express an hCD147 transgene at heterozygous or homozygous levels (Supplementary Fig. 17). Specifically, a human cDNA encoding CD147 was targeted to mouse CD147 exon 1. The resulting knock-in created a fusion protein composed of the first 22 amino acids of the mouse CD147 signal peptide and amino acids 23–385 of the hCD147 (NP_001719.2) expressed under the control of the endogenous mouse CD147 promoter. Transcription termination was mediated by a bovine growth hormone polyadenylation signal sequence. Targeting was performed directly in NSG mouse embryos (JAX stock# 005557) by coinjecting a targeting vector and Cas9 protein complexed with a CRISPR sgRNA recognizing and cutting the sequence 5′-GCCTGCGCGGCGGGTAAGAG-3′. The targeted alleles were subsequently sequenced in their entirety by PCR genotyping, and 14 positive founders were correctly targeted.

**Statistical analysis.** Tumor size statistical analysis was performed by two-way ANOVA (analysis of variance) with Bonferroni post tests. The overall survival statistics were calculated using the log-rank test. Other statistical significance was determined using a two-tailed unpaired Student's $t$ test or a two-tailed paired Student's $t$ test. All statistical calculation graphs were generated by GraphPad

Prism 5.0. $P < 0.05$ (*), $P < 0.01$(**), and $P < 0.001$(***) were considered statistically significant.

## Data availability

Patient survival data and RNA-seq by expectation maximization normalized expression datasets were downloaded from OncoLnc [http://www.oncolnc.org/] or from FireHose Broad GDAC [https://gdac.broadinstitute.org/]. All remaining relevant data are available in the article, Supplementary information, or from the corresponding author upon reasonable request. Source data are provided with this paper.

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

## Acknowledgements

This work was supported, in part, from HL125018 (D.L.), AI124769-01 (D.L.), AI129594 (D.L.), AI130197 (D.L.), and Rutgers University-New Jersey Medical School Liu Laboratory Startup funding. We thank Dr. Rongfu Wang (Houston Methodist Research Institute) for providing FFLuc-GFP plasmids. We highly appreciate the comments of this manuscript and support by Drs. Ke Geng and Celso Viana (Rutgers University) and other members from Dr. D.L.'s laboratory (Drs. Xuening Wang, Chih-Hsiung Chen, Shuo Tian, and Yi-Hsing Hsu). We also would like to thank Dr. Eric Long (NIAID/NIH) for the 721.221 cell line.

## Author contributions

D.L. directed the project, designed and performed experiments, interpreted data, and wrote the manuscript. H.-C.T., W.X., Y.Y., T.L., S.B., Q.Y., M.M., and L.F. performed experiments and contributed to the preparation of the manuscript. C.A.R., G.D., J.G., J.-G.J., W.-X.Z., and C.L. contributed to the preparation of manuscript.

## Competing interests

The authors declare no competing interests.
