## [Peer Review File · Nature Communications]

Reviewers' comments:

Reviewer #1 (Remarks to the Author):

This is a significantly revised manuscript to evaluate the efficacy of anti-CD147 CARs to target hepatocellular carcinoma using both CAR-expressing T cells as well as CAR-expressing NK cells. The authors provide an extensive rebuttal and revised manuscript from the previous version. While many issues were addressed there remain some issues and inconsistencies in the manuscripts. Most generally, the emphasis now is on the suitability to target CD147 using either CAR-NK or CAR-T cells, rather than just focusing on CAR-NK cells in the previous version. Some studies also now use a Syn-Notch CAR-T cell system combining CD147 and GCP-3 targeting that may provide a safer approach. However, in testing all these multiple systems the data can be hard to follow and the overall focus of the work gets lost. As noted, there remain issues to be addressed to make more suitable.

Major points:

1. One of the key issues in the previous reviews was the use of unirradiated NK92 cells whereas clinical studies must be done with irradiated NK92 cells. These studies now use and compare both irradiated and non-irradiated cells such as studies in the new Supplemental Figure 11 that remarkably shows better antitumor activity with the irradiated NK92 cells. However, when comparing studies with NK92 cells to those that use primary T or NK cells, the methods and timing to administering these cells are different from experiment to experiment. For example, in Figures 3 and Supplemental Figure 11 use doses of cells on days 5, 7, 9, 16, and 18, whereas other studies as in Figure 4 give injections starting on day 1, in one study injections go on for 5 days in another study they go on for 29 days. It is not clear why these are different and whether these experimental models were predefined. In order to best compare efficacy of all these different cell populations in targeting CD147 positive tumors, it would be best to let the tumor establish for 4-5 days and then give just one injection of the CAR modified T cells or NK cells, or at a minimum, have the same experimental design in all in vivo studies unless clear reason to vary. Indeed, this is what is done in previous studies using primary (cord blood) CAR-NK cells or iPSC-derived NK cells such as the work referred to in the reply using these different cell populations (see, *N Engl J Med* 382, 545-553 (2020) and *Cell Stem Cell* 23, 181-192 e185 (2018)).

2. Another concern in the previous reviews was persistence of the anti-CD147 CAR-NK cells, especially when using irradiated NK92 cells. The authors reply that addressing this question would require a separate research project. In reality, it is quite easy to determine persistence of these cells by bleeding the mice during the in vivo studies conducted. Instead, the authors demonstrate studies using primary anti-CD147 CAR-NK cells and CAR-T cells in their Figure R4. They have not done this with NK92 cells. Surprisingly there are really no CAR-NK cells or CAR-T cells seen in the blood of these mice after only four days when you would typically expect to see cells. One figure does show CD56+ cells in the spleen of NK cell treated mice, though surprisingly there is also a substantial population of human CD3+ cells which suggests that the sorting or selection of the primary NK cells may not have been that specific.

3. Another concern addressed in the reply is expression of TIM3 and other exhaustion markers on the NK cell populations. The authors demonstrate in Figure R1 that the CD147 CAR-NK cells have low expression of TIM3 and again it is surprising that this is different than the wild-type NK92 cells that have higher levels of expression. Similarly, the CD147 CAR-NK cells from peripheral blood have high levels of Tim3, perhaps as a result of the manufacturing process that requires extensive cell proliferation. This Tim3 expression is also surprising as these cells seem to be as effective in

the in vivo studies, though expression of these exhaustion markers is of concern is the primary NK cells were to be used for clinical studies.

4. Another concern in the previous reviews was expression of CD147 on normal cells and tissues. The authors show that two B cell tumor lines, Raji and Daudi, express high levels of CD147. As the authors note that CD147 is found on normal hematopoietic cells, it would be useful to know if this is found on normal B cells whether this would be a problem for clinical translation, either due to off target effects or inhibiting the efficacy for clinical use due to competition from the normal B cells or other CD147 positive expressing cell.

5. The authors also note that NK92 cells express CD147 though surprisingly this does not inhibit the production and growth of anti-CD147 CAR-NK92 cells, suggesting that this expression does not lead to fratricide that would inhibit the production of anti-CD147 CAR-NK92 cells. More data about this would be useful, and it would also be useful to know if normal NK cells and/or primary T cells also express CD147.

Reviewer #4 (Remarks to the Author):

There are significant problems with the in vivo studies, mainly because the anti-CD147 CAR does not recognize the mouse protein. As a result, it is unclear what would be the level of toxicity if indeed it recognized the mouse protein. Therefore, it is unclear whether the strategy is going to be useful at all. Also, the mouse model does not address tumor environment of HCC.

Point-by-point reply to all reviewers' comments regarding manuscript NCOMMS-20-07002-T by Xiong, W., et al., 'Efficacy of anti-CD147 Chimeric Antigen Receptors Targeting Hepatocellular Carcinoma'.

General comments:

This manuscript has been revised extensively to address all points raised during the second review. Where suggested, additional experiments have been performed and included in the second revised manuscript. We revised the manuscript thoroughly. We also provided additional data (including the hCD147TG mouse model) in the second revision.

There are 7 new figures, including four new figures (New Figures R2-1, R2-2, R2-3, and R2-4). To enhance the readability and focus of the manuscript, these 4 new figures have been added in response to this point-by-point reply to Reviewer file here. In addition to 4 new figures in this point-by-point reply to Reviewer file, three new Supplementary figures (Supplementary Figure S16, S17, and S18) have been included in the second revised manuscript.

All changes we made have been marked with Track Changes highlighting in a separate file.

Reviewer #1 (Remarks to the Author):

This is a significantly revised manuscript to evaluate the efficacy of anti-CD147 CARs to target hepatocellular carcinoma using both CAR-expressing T cells as well as CAR-expressing NK cells. The authors provide an extensive rebuttal and revised manuscript from the previous version. While many issues were addressed there remain some issues and inconsistencies in the manuscripts. Most generally, the emphasis now is on the suitability to target CD147 using either CAR-NK or CAR-T cells, rather than just focusing on CAR-NK cells in the previous version. Some studies also now use a Syn-Notch CAR-T cell system combining CD147 and GCP-3 targeting that may provide a safer approach. However, in testing all these multiple systems the data can be hard to follow and the overall focus of the work gets lost. As noted, there remain issues to be addressed to make more suitable.

Response: We thank the Reviewer #1 for the positive comments. We have re-organized the data and data presentation. We have clarified the different approaches and systems in the revision. To enhance scientific rigor and reproducibility required by NIH¹ and Nature Communications², we repeated our experiments using multiple systems. For example, to demonstrate CD147-CAR-NK cells controlling progression of HCC *in vivo*, we used four different *in vivo* systems: **First**, we used CD147-CAR-modified NK-92MI cells without irradiation (Figure 4); **Second**, we used CD147-CAR-modified NK-92MI with irradiation and compared the efficacy between irradiated NK-92MI and non-irradiated NK-92MI (Supplementary Figure S11); **Third**, we used CD147-CAR-modified primary NK cells (Figure 5). **Fourth**, we used both a xenograft NSG mouse model and a patient-derived xenograft mouse model (Figure 4e).

We believe that the set of NK alone data (without including CD147-CAR-T cell data) can be an individual manuscript. We would like to emphasize a point that the data and scientific discoveries presented in the current manuscript can truly demonstrate the scientific rigor and reproducibility. We greatly appreciate the reviewers' constructive comments and revised the manuscript in the second revision.

Major points:

1. One of the key issues in the previous reviews was the use of unirradiated NK92 cells whereas clinical studies must be done with irradiated NK92 cells. These studies now use and compare both irradiated and non-irradiated cells such as studies in the new Supplemental Figure 11 that remarkably shows better antitumor activity with the irradiated NK92 cells. However, when comparing studies with NK92 cells to those that use primary T or NK cells, the methods and timing to administering these cells are different from experiment to experiment. For example, in Figures 3 and Supplemental Figure 11 use doses of cells on days 5, 7, 9, 16, and 18, whereas other studies as in Figure 4 give injections starting on day 1, in one study injections go on for 5 days in another study they go on for 29 days. It is not clear why these are different and whether these experimental models were predefined. In order to best compare efficacy of all these different cell populations in targeting CD147positive tumors, it would be best to let the tumor establish for 4-5 days and then give just one injection of the CAR modified T cells or NK cells, or at a minimum, have the same experimental design in all in vivo studies unless clear reason to vary. Indeed, this is what is done in previous studies using primary (cord blood) CAR-NK cells or iPSC-derived NK cells such as the work referred to in the reply using these different cell populations (see, N Engl J Med 382, 545-553 (2020) and Cell Stem Cell 23, 181-192 e185 (2018)).

Response: We thank the reviewer #1 for the positive comments. We realized the different definitions of "day 0" among different experiments in the different figure legends. We understand that it is confusing. We have corrected this in the revised figure legends.

We appreciate that the reviewer's comments about injection after 4-5 days tumor implantation. We performed the assays as the reviewer's suggestion. The description among different figure legends is different, which is confusing. In fact, all our treatment regimens started at tumor size around 50 mm³ (formula Volume [in cubic millimeters] = $[L \times (W)^2]/2$ [W, width; L, length]), which is common in the field of cancer research^{3,4}. It usually takes 5 to 7 days, as described in the "Materials and Methods". The "day 0" in Figure S11 and Figure 3 was defined as the "initial tumor cell injection time point". However, the "day 0" in Figure 4 was defined as "the day before CAR treatment". We are sorry for the inconsistency in definition of "Day 0". We have corrected this in the revision.

Of note, due to variability in batch-to-batch tumor growth curves *in vivo*, it is quite difficult to maintain a standard dosing regimen between experiments from different researchers (which further demonstrated the scientific rigor and reproducibility). Additionally, as CAR-modified

cells derived from different sources show slightly different tumor killing capabilities *in vitro*, we expect there to be significant differences *in vivo* as well. Our goals in this study was not to systematically determine which type of CAR-modified therapy was more effective at tumor growth control, but such a study would definitely be valuable to the field of immunotherapy. We extended the dosing regimen in some studies to determine if tumor control was an early event or a characteristic of long-term therapy as well. Importantly, there are several groups who evaluated the therapeutic potential of CAR therapy after multiple infusions of engineered immune cells^{5,6,7}. Thus, it is feasible to use multiple doses for NK cells and in immunocompromised HCC patients.

In summary, we would like to point out the following rationales: **1)** variability in batch-to batch tumor growth *in vivo*; **2)** variability of CAR-modified cells derived from different sources; **3)** different experience level of different researchers. Different judgement of experimental observation (e.g., variable tumor size measurements) from different experimenters; **4)** In clinical trials and clinical practice, CAR dosing is variable, which depends on experimental design from different physicians and scientists. In our experiments, we minimized the discrepancy among different batch-to-batch tumors, different sources of CAR modified cells, and different experimenters, by detecting the size of tumors (around 50 mm³).

2. Another concern in the previous reviews was persistence of the anti-CD147 CAR-NK cells, especially when using irradiated NK92 cells. The authors reply that addressing this question would require a separate research project. In reality, it is quite easy to determine persistence of these cells by bleeding the mice during the in vivo studies conducted. Instead, the authors demonstrate studies using primary anti-CD147 CAR-NK cells and CAR-T cells in their Figure R4. They have not done this with NK92 cells. Surprisingly there are really no CAR-NK cells or CAR-T cells seen in the blood of these mice after only four days when you would typically expect to see cells. One figure does show CD56+ cells in the spleen of NK cell treated mice, though surprisingly there is also a substantial population of human CD3+ cells which suggests that the sorting or selection of the primary NK cells may not have been that specific.

Response: We thank the Reviewer #1 for the evaluation. We agree with the Reviewer #1. The detailed analysis of CAR-NK persistence *in vivo* and comparison between CAR-NK persistence and CAR-T persistence *in vivo* is an important question. In the field of immunotherapy, it is still disputable because it is critical for clinical efficacy, response durations, toxicity, and other clinical applications. To thoroughly address the persistence of anti-CD147-CAR-NK cells, we are designing a separate research project on this topic.

We addressed the Reviewer #1's critiques by carrying out a series of different assays as follows:

To address the reviewers' concerns, we performed a pilot assay to evaluate the persistence and survival capability of primary CD147-CAR-NK and CD147-CAR-T cells in mouse blood and spleen

(the old Figure R4 in the point-by-point reply to reviewers file). We provided this data in the old Figure R4.

Consistent with the Reviewer #1's interpretation, we agreed that it is surprising that there are very limited CAR-NK and CAR-T cells in blood. We honestly and carefully presented this data. Thus, we hypothesize that the CAR-modified immune cells (including T and NK cells) may migrate into spleen, tumor tissue, or other organs (e.g., bone marrow and liver).

Indeed, we showed the increased percentage of NK cells in spleen, compared to the percentage of NK cells in blood (old Figure R4). We also quantified the length of the spleen and compared the spleen length among different groups, we observed that the spleen length on CD147-CAR-NK-treated group (the new Figure R2-1) significantly increased, compared with PBS control and CAR-T treated groups, which is consistent with the data presented in the previous Figure R4.

Figure R2-1: Increased spleen length from mice treated with primary CD147-CAR-NK. (A) Representative spleen sizes from different groups treated with different CAR cells and control groups. (B) Increased spleen length indicates the accumulation of CD147-CAR-NK in the spleen after CD147-CAR-NK injection, compared with control groups and CD147-CAR-T treated group. SK-Hep-1-bearing mice were injected (*i.v.*) with expanded primary NK cells, CD147-CAR primary NK cells, and CD147-CAR primary T cells. The spleen from each individual mouse was collected at different time points due to the different endpoints for Kaplan-Meier survival curves. Data are representative of two independent experiments. The scale bars represent 10 mm.

In addition to the length of spleen measurements, we also examined the persistence of injected NK cells by immunohistochemical (IHC) stain on liver and tumor tissues. As expected, we

observed the persistence of primary NK/CAR-NK cell and accumulation of primary NK/CAR-NK cells in the liver and tumor tissue after 58- to 63-day injection (Figure R2-2). Briefly, Mice injected with SK-Hep1 hepatocellular carcinoma cells subcutaneously were treated either with wild-type expanded NK cells, CD147-CAR NK cells, or with PBS. At the experimental endpoint (tumor volume > 2000 mm³), mice were euthanized, and their liver and tumors were fixed in 4% PFA. Slides were prepared and stained anti-human CD56 (Clone: 123C3; 1:300 dilution; pH 9 antigen retrieval solution). Arrows in the Figure R2-2 indicate positive staining of NK cells within the primary tumor and mouse liver afflicted with tumor metastasis.

Figure R2-2: Persistence and tissue distribution of CAR-NK cells from mice treated with primary CD147-CAR-NK. Experimental design with three different groups is demonstrated in the left panel. Briefly, mice injected subcutaneously with SK-Hep1 hepatocellular carcinoma cells were treated either with wild-type expanded NK cells (Group #1: NT NK cells), CD147-CAR-NK cells (Group #2), or with PBS (Group #3). At the experimental endpoint (tumor volume > 2000 mm³, about 58-63 days after CAR-NK injection), mice were euthanized. And their liver and tumor tissues were fixed in 4% PFA. Slides were prepared and stained anti-human CD56 (Clone 123C3; 1:300 dilution). Arrows indicate positive staining of NK cells within the primary tumor and mouse liver afflicted with tumor metastasis. The scale bars represent 200 μm.

Compared to CAR-NK cells, there are few numbers of CD147-CAR-T cells in both blood and spleen. We hypothesize that these CD147-CAR-T cells may accumulate into tumor tissue or other organs. Additional experiments have been designed to address this question.

Finally, we agreed with the Reviewer #1's observation and interpretation. In Figure R4, there are 0.3% of CD3 positive T cells because of the CD147-CAR-NK

derived from PBMCs. Because these CD3⁺ T cells were gated on live singlets (NOT CAR positive T cells), we believe that the percentage of CAR⁺ CD3⁺ T cells could be lower than 0.3%, which is a reasonable or acceptable contamination (e.g., 0.3%, less than 1%) from T cells during cell expansion and purification. Therefore, the efficacy of CAR-NK cells in our experiments is valid. Nevertheless, we are currently optimizing our protocol to increase the purity of CAR-NK cells.

3. Another concern addressed in the reply is expression of TIM3 and other exhaustion markers on the NK cell populations. The authors demonstrate in Figure R1 that the CD147 CAR-NK cells have low expression of TIM3 and again it is surprising that this is different than the wild-type NK92 cells that have higher levels of expression. Similarly, the CD147 CAR-NK cells from peripheral blood have high levels of Tim3, perhaps as a result of the manufacturing process that requires extensive cell proliferation. This Tim3 expression is also surprising as these cells seem to be as effective in the in vivo studies, though

expression of these exhaustion markers is of concern is the primary NK cells were to be used for clinical studies.

Response: We thank the Reviewer #1 for the comments. We do appreciate the Reviewer's concern on increased TIM3 expression in primary CD147-CAR-NK cells, but decreased TIM3 expression CD147-CAR-NK-92MI. The molecular mechanism(s) underlying the TIM3 difference needs further investigation in the future. Considering the increased TIM3 expression on primary CD147-CAR-NK cells, we will use CRISPR-Cas9 KO technology to delete the TIM3 molecule on exhausted CD147-CAR-NK cells in the future, as described^{8,9,10,11}.

We have clarified this in the discussion section.

4. Another concern in the previous reviews was expression of CD147 on normal cells and tissues. The authors show that two B cell tumor lines, Raji and Daudi, express high levels of CD147. As the authors note that CD147 is found on normal hematopoietic cells, it would be useful to know if this is found on normal B cells whether this would be a problem for clinical translation, either due to off target effects or inhibiting the efficacy for clinical use due to competition from the normal B cells or other CD147 positive expressing cell.

Response: We thank the Reviewer #1 for the comments.

We analyzed the CD147 RNA expression on B cells by analyzing the publicly available databases. The CD147 expression on B cells from peripheral blood is minimal. Again, in the previous rebuttal letter and main text, we explain that we have optimized a single chain variable fragments (scFv) sequence that binds to only tumor cells with a high surface expression level of the CD147 antigen. This sequence for the scFv domain is unique. The modified scFv that specifically binds CD147 is provided in this study. Secondly, to further alleviate the "on-target/off-tumor" toxicity concern, we provide new data about GPC-3-CAR in combination with CD147-SynNotch modified CAR T cell data, as well as the CD147-CAR in combination with GPC-3-SynNotch modified CAR T data (new **Figure 6, Supplementary Figure S13, S14 and S15**).

To further address the concern about CD147 expression on B cells, at the protein level, we measured the expression of CD147 on mixed PBMC's (Supplementary Figure S12). The CD147

Figure R2-3: CD147 does not express on human primary B cells isolated from PBMCs. Representative histograms of the expression of CD147 on primary CD19⁺ B cells freshly isolated from three healthy donors' PBMCs, respectively. The MFI is noted in the respective histograms. Data are representative of two independent experiments.

expression on PBMCs is very low. We have provided this data in Figure S1c and Figure S12 in the first revision.

As requested by Reviewer #1, to specifically interrogate the CD147 expression level on B cells, we stained B cells with CD19 and CD147. We noticed that the total CD19+ population does not express CD147. (**Figure R2-3**)

Additionally, we also performed the bioinformatical analysis on different subsets of B cells from the following public databases:

<https://www.proteinatlas.org/>

<https://portals.broadinstitute.org/ccle>

<http://biogps.org/#goto=welcome>

<https://www.gtexportal.org/home>

<https://fantom.gsc.riken.jp/zenbu/>

We quantified BSG (also known as CD147) expression as FPKM (Fragments Per Kilobase of gene model per Million reads per sample) from these public databases. Consistent with our flow data, there is undetectable CD147 mRNA presence on healthy B cells (data not shown), compared with cancer cells. Again, the scFv of CD147 we chose selectively binds cells with high level of CD147 expression on their surface. Thus, we believe that CD147 expression will not inhibit the efficacy for clinical use of CD147-CAR.

5. The authors also note that NK92 cells express CD147 though surprisingly this does not inhibit the production and growth of anti-CD147 CAR-NK92 cells, suggesting that this expression does not lead to fratricide that would inhibit the production of anti-CD147 CAR-NK92 cells. More data about this would be useful, and it would also be useful to know if normal NK cells and/or primary T cells also express CD147.

Response: We thank the Reviewer #1 for the comment. We noticed that the low expression of CD147 molecules on NK-92MI cell lines and provided data in Figure S4. We hypothesize that limited fratricide among CD147-CAR-NK-92MI during expansion occurs and selectively expands the CD147 negative population or downregulation of CD147 molecules during NK-92MI fratricide occurring at the beginning of transduced cell expansion. After 7-day expansion, the majority of CD147-CAR-NK-92MI became CD147 negative or expressed extremely low levels of CD147 (**Figure R2-4**). We have clarified this in the revised manuscript.

Notably, the loss (or down-regulation) of CD147 molecule expression on CD147-CAR-NK-92MI cells did not affect their functionalities and expansion *in vitro* and *in vivo*. In contrast, CD147-CAR-NK-92MI cells with CD147-knockout (a separate NK-92MI cell line with CD147-knockout prior to CAR transduction) show an increased cytotoxicity against their susceptible target cells and cytokine production (data not shown, because the role of CD147 in NK cells is beyond the scope of [and would distract from] the current manuscript.).

Additionally, we have optimized our scFV sequence for selectively binding with highly-expressed CD147 antigen. Since CD147 is also expressed, albeit at extremely low levels, on some healthy tissues, this optimized scFV cannot bind with low levels of CD147 (**Figure S12** in the first revised manuscript).

Normal NK and primary T cells express extremely low level of CD147 molecules. However, upon stimulation by activating signals (such as susceptible target cells), CD147 expression levels on 5-10% of T and NK cells increase slightly. The majority of NK and T cells remains CD147 negative. After generation of CD147-CAR-primary NK and T cells, these up-regulated CD147 primary T and NK cells will be eliminated by CD147-CAR cells via limited, transient fratricide (**Figure R2-4**). The remaining CD147 negative or CD147 low, but CD147-CAR positive cells can be expanded and further proliferate.

Figure R2-4: Limited and transient fratricide promotes CD147 negative CD147-CAR-NK cell expansion.

Representative diagrams show the transient fratricide of CD147 positive NK cells during expansion. Initially, there are two distinct CD147 positive and CD147 negative/low NK populations (**left bottom panel**). After CAR transduction and stimulation, CD147 molecules are upregulated on one subset of NK cells. The upregulated CD147 molecules triggered fratricide by neighboring CD147-CAR-NK cells during CAR-NK expansion, which significantly reduced the number of CD147 positive NK or CAR-NK cells. The remaining CD147 negative CAR-NK or CD147^{low} CAR-NK cells can be further expanded. Thus, transient fratricide promotes the CD147 antigen negative or CD147^{low} CAR-NK cell expansion and function.

We have clarified this point in the revision.

Reviewer #4 (Remarks to the Author):

There are significant problems with the in vivo studies, mainly because the anti-CD147 CAR does not recognize the mouse protein. As a result, it is unclear what would be the level of toxicity if indeed it

recognized the mouse protein. Therefore, it is unclear whether the strategy is going to be useful at all. Also, the mouse model does not address tumor environment of HCC.

Response: We thank the Reviewer #4 for the comments. To address the level of toxicity, we developed 'logic-gated' CD147-CAR in the revision. We provided new data regarding GPC-3-CAR in combination with CD147-SynNotch modified CAR T cells, as well as the strategy involving CD147-CAR in combination with GPC-3-SynNotch modified CAR T cells (new **Figure 6** and Supplementary **Figure S13-S15** in the first revised manuscript). Thus, we have "logic-gated" CAR-T and CAR-NK cells to address the toxicity of CD147-CAR in the revised manuscript.

We have provided new data using a HepG2 HCC tumor cell line xenograft *in vivo* NSG mouse model in the Supplementary Figure **S16**. The new data show that 'logic-gated' CD147-CAR is effective. No obvious toxicity (measured by Body Weight and Appearance/Activity) has been observed.

Furthermore, we also tested a pilot assay to test the toxicity of the GPC3-synNotch inducible CD147-CAR-NK-92MI (logCD147-CAR) cells in human CD147-transgenic (hCD147TG) NSG mice. Specifically, to generate a hCD147TG mouse, a human cDNA encoding CD147 was targeted to mouse CD147 exon 1. The resulting knock-in created a fusion protein composed of the first 22 amino acids of the mouse CD147 signal peptide and amino acids 23-385 of the human CD147 (NP_001719.2) expressed under the control of the endogenous mouse CD147 promoter. Transcription termination was mediated by a bovine growth hormone polyadenylation signal sequence. Targeting was performed directly in NSG (JAX stock# 005557) background mouse embryos by co-injecting a targeting vector and Cas9 protein complexed with a CRISPR sgRNA recognizing and cutting the sequence 5'-GCCTGCGCGGGTAAGAG-3'. The targeted alleles were subsequently sequenced in their entirety by PCR genotyping, and 14 positive founders were correctly targeted. Genotyping confirmed the successful generation of human CD147-transgenic (Supplementary Figure **S17A**). IHC and Flow cytometry analysis showed human CD147 expression on liver tissues on hCD147TG mice (Supplementary Figure **S17B**). The genotyping products were verified by sequencing (data not shown).

To evaluate the toxicity of "logic-gated" GPC3-synNotch inducible CD147-CAR-NK-92MI (logCD147-CAR) cells in the hCD147TG mouse model, we primed the GPC3-synNotch inducible CD147-CAR-NK-92MI with irradiated, CD147^{high} GPC3^{high} HepG2 (dual-antigen positive) cells overnight. The expression of CD147-CAR was verified by flow cytometry (Supplementary Figure **S18**). Then, the primed GPC3-synNotch inducible CD147-CAR-NK-92MI cells were irradiated, and injected into the hCD147TG mouse. After three doses of primed, irradiated GPC3-synNotch inducible CD147-CAR-NK-92MI cell injections, both acute wasting syndrome (e.g., body weight) and survival within 4 to 8 days post-injection with three doses have been measured

(Supplementary Figure **S18**). As expected, the body weight (Appearance and activity were also recorded.) of treated hCD147TG mouse did not decrease, indicating the tolerable toxicity in hCD147TG mouse model.

We understand that the hCD147TG mouse model does not really address the safety/toxicity issue in humans. What we observed in the hCD147TG mouse model may not be the same in humans. The true safety of 'off-target' toxicity using CD147-CAR-T/NK or logCD147-CAR-T/NK cells needs to be further tested in future clinical trial studies. It is not fair for us to address all the related questions in one single manuscript.

Since we cannot perform the orthotopic model system to address the tumor microenvironment of HCC in this study, we have highlighted the limitations of our study and the safety/toxicity caveat in our conclusions.

References:

1. Collins FS, Tabak LA. Policy: NIH plans to enhance reproducibility. *Nature* **505**, 612-613 (2014).
2. Reproducibility: let's get it right from the start. *Nat Commun* **9**, 3716 (2018).
3. Ciamporcero E, *et al.* Combination strategy targeting VEGF and HGF/c-met in human renal cell carcinoma models. *Mol Cancer Ther* **14**, 101-110 (2015).
4. Jensen MM, Jorgensen JT, Binderup T, Kjaer A. Tumor volume in subcutaneous mouse xenografts measured by microCT is more accurate and reproducible than determined by 18F-FDG-microPET or external caliper. *BMC Med Imaging* **8**, 16 (2008).
5. He Y, Li XM, Yin CH, Wu YM. Killing cervical cancer cells by specific chimeric antigen receptor-modified T cells. *J Reprod Immunol* **139**, 103115 (2020).
6. Zhao Y, *et al.* Multiple injections of electroporated autologous T cells expressing a chimeric antigen receptor mediate regression of human disseminated tumor. *Cancer Res* **70**, 9053-9061 (2010).
7. Globerson-Levin A, Waks T, Eshhar Z. Elimination of progressive mammary cancer by repeated administrations of chimeric antigen receptor-modified T cells. *Mol Ther* **22**, 1029-1038 (2014).
8. Ren J, Liu X, Fang C, Jiang S, June CH, Zhao Y. Multiplex Genome Editing to Generate Universal CAR T Cells Resistant to PD1 Inhibition. *Clin Cancer Res* **23**, 2255-2266 (2017).
9. Rupp LJ, *et al.* CRISPR/Cas9-mediated PD-1 disruption enhances anti-tumor efficacy of human chimeric antigen receptor T cells. *Sci Rep* **7**, 737 (2017).
10. Choi BD, *et al.* CRISPR-Cas9 disruption of PD-1 enhances activity of universal EGFRvIII CAR T cells in a preclinical model of human glioblastoma. *J Immunother Cancer* **7**, 304 (2019).

11. Hu W, *et al.* CRISPR/Cas9-mediated PD-1 disruption enhances human mesothelin-targeted CAR T cell effector functions. *Cancer Immunol Immunother* **68**, 365-377 (2019).

REVIEWERS' COMMENTS:

Reviewer #1 (Remarks to the Author):

Overall, this work is improved with the additional revisions. However, there remain a couple issues:

- The utility of the syn-Notch studies is unclear. The data in Figure S16 only shows a minimal benefit in killing the GPC-high compared to GPC-low tumor cells at one time point (D. 18) with no longer data shown. What happened after this time point? Is there a difference in survival of the mice? While the in vitro studies are reasonable- this is an approach already demonstrated by other groups and similar tests have not been done in vivo to show specificity of this approach.
- For studies in Figure 4 that continue to use NK92 cells, please indicate if the NK cells have been irradiated or not. As noted before, tests with unirradiated NK92 cells are not particularly valuable.
- Studies of NK cell (and T cell) persistence remain confusing and not included in the main manuscript. The authors now show that mice treated with NK cells, but not T cells have larger spleens. However, they do not demonstrate the spleens contain the human NK cells. Also, the tumor staining shows minimal infiltration of human NK cells and not clear if a difference between infiltrates in mice that receive CAR+ vs CAR- NK cells. Therefore, if the NK cells are not seen in the blood, spleen or tumor, where do the authors think these cells are and how do the CAR+ NK cells mediate improved anti-tumor activity?

Reviewer #4 (Remarks to the Author):

No new comment. The authors have somewhat addressed my major concern on toxicity by generating a new mouse strain that expresses the human CD147.

Point-by-point reply to all reviewers' comments regarding manuscript NCOMMS-20-07002-T by Tseng, H.C., et al., '**Efficacy of anti-CD147 Chimeric Antigen Receptors Targeting Hepatocellular Carcinoma**'.

General comments:

This manuscript have been further modified according to Editor and Reviewers' comments. All changes we made have been marked with Track Changes highlighting in a separate file.

Reviewer #1

Overall, this work is improved with the additional revisions. However, there remain a couple issues:

Response: We thank the Reviewer #1 for the positive comments.

- The utility of the syn-Notch studies is unclear. The data in Figure S16 only shows a minimal benefit in killing the GPC-high compared to GPC-low tumor cells at one time point (D. 18) with no longer data shown. What happened after this time point? Is there a difference in survival of the mice? While the in vitro studies are reasonable- this is an approach already demonstrated by other groups and similar tests have not been done in vivo to show specificity of this approach.

Response: We thank the Reviewer #1 for the constructive comments. Although these findings in this manuscript support the therapeutic potential of CD147-CAR cells for treating HCC patients (one of deadliest solid cancers in humans), there are several limitations presented in the current form of study. First, we did not provide the difference in survival of the mice between synNotch-CAR treated group and control groups. We will incorporate the suggested data in our future studies, but current laboratory conditions, including COVID-19 restrictions, preclude the performance of these experiments. The non-COVID-19 related research has been temporarily suspended in most research institutions due to the ongoing COVID-19 pandemic. Second, we will state the limitations in the main text. Third, the main focus of this paper is to demonstrate the efficacy of targeting CD147 for HCC. We will continue this research after COVID-19.

- For studies in Figure 4 that continue to use NK92 cells, please indicate if the NK cells have been irradiated or not. As noted before, tests with unirradiated NK92 cells are not particularly valuable.

Response: We thank the Reviewer #1 for the positive comments. We have clarified this in the revised main text.

- Studies of NK cell (and T cell) persistence remain confusing and not included in the main manuscript. The authors now show that mice treated with NK cells, but not T cells have larger spleens. However, they do not demonstrate the spleens contain the human NK cells. Also, the

tumor staining shows minimal infiltration of human Nk cells and not clear if a difference between infiltrates in mice that receive CAR+ vs CAR- NK cells. Therefore, if the NK cells are not seen in the blood, spleen or tumor, where do the authors think these cells are and how do the CAR+ NK cells mediate improved anti-tumor activity?

Response: We thank the Reviewer #1 for the constructive comments. Now, we have provided new data about NK infiltration in both spleen and liver tissue. We agree with the Reviewer #1 that it is useful to show the actual NK infiltration in the spleen by IHC assays.

Figure R3-1: Persistence and tissue distribution of CAR-NK cells from mice treated with primary CD147-CAR-NK.

Experimental design with three different groups is demonstrated in the left panel. Briefly, mice injected subcutaneously with SK-Hep1 hepatocellular carcinoma cells were treated either with wild-type expanded NK cells (Group #1: NT NK cells), CD147-CAR-NK cells (Group #2), or with PBS (Group #3). At the experimental endpoint (tumor volume > 2000 mm³, about 58-63 days after CAR-NK injection), mice were euthanized. And their liver (top), tumor (middle), and spleen (bottom) tissues were fixed in 4% PFA. Slides were prepared and stained with anti-human CD56 (Clone 123C3; 1:300 dilution). Arrows indicate positive staining of NK cells within the primary tumor and spleen, as well as within mouse livers afflicted with tumor metastases. The scale bars represent 200 μm.

We did observe CAR-NK cell infiltration in the spleen, liver, and tumor tissues. We see a very low percentage of NK cells in the blood because of possible infiltration of NK cells into liver and spleen. The CAR-NK staining in tumor, liver, and spleen was performed at the experimental end points (Figure R3-1), which explains why non-massive infiltration of CAR-NK was observed in the IHC data.

In addition to the data provided in the Figure R2-1A, the significantly increased length of spleens in mice treated with NK cells was observed (Figure R2-1B) in the previous ‘Reply to Reviewers’ file.

In addition to the NK infiltration in spleen tissue

IHC assay (Figure R2) in the previous ‘Reply to Reviewers’ file, we also examined the liver weight and size. As shown in Figure R3-2 in the current ‘Reply to Reviewers’ file, we provided new data about the liver weight on the group treated with CD147-CAR-NK cells.

Unexpectedly, we observed slightly increased liver weight after 58- to 63-days following injection of CAR-modified immune cells (Figure R3-2). Briefly, mice injected with SK-Hep1 hepatocellular carcinoma cells subcutaneously were treated either with wild-type expanded NK

(Non-transduced NK cells [NT-NK]) cells, CD147-CAR-NK cells, or with PBS. At the experimental endpoint (tumor volume > 2000 mm³), mice were euthanized, and their liver tissues were fixed in 4% PFA.

Interestingly, we observed that the liver weight of mice from the CD147-CAR-NK treated group is slightly higher than that of the CD147-CAR-T treated group, although these differences were not able to reach statistical significance ($p = 0.06$) due to the relatively small sample size ($n = 10$) and experimental endpoint studies.

The molecular mechanisms for why CD147-CAR preferentially infiltrated liver remains unclear. We propose three possible potential mechanisms:

(1) NK cells are highly enriched in human liver. The percentage of NK cells in total intrahepatic lymphocytes in human and mouse livers are 25-40% and 10-20%, respectively¹. The injected CAR-NK cells have an intrinsic capacity to accumulate into liver tissue. The percentage of intrahepatic NK cells in the liver is around 5 times higher than the percentage of peripheral blood NK cells in humans^{2,3}.

Figure R3-2: Increased liver size and weight from mice treated with primary CD147-CAR-NK, compared to the group treated with CD147-CAR-T cells. (A) Representative liver sizes from different groups treated with various CAR cells and control groups. **(B)** Increased liver weight indicates the accumulation of CD147-CAR-NK in the liver after CD147-CAR-NK injection, compared with control groups and CD147-CAR-T treated group. SK-Hep-1-bearing mice were injected (*i.v.*) with expanded primary NK cells, CD147-CAR primary NK cells, and CD147-CAR primary T cells. The liver from each individual mouse was collected at different time points due to different experimental endpoints for generating Kaplan-Meier survival curves. Data are from one experiment. The stated scales are indicated by a ruler.

(2) NK cells highly express integrin (CD11a and CD11b) and inflammatory chemokine receptors (CXCR1 and CX3CR1), compared to effective CD8+ cytotoxic T cells⁴. These highly expressed integrin and chemokine receptors preferentially recruit NK cells in the liver.

(3) The liver is the largest solid organ of the human body, which is responsible for the removal of the pathogens such as bacteria from the systemic circulation⁵. NK cells can quickly provide the first line of defense in the liver tissue, which explains the highly enriched NK lymphocyte populations in the liver.

In summary, we provide additional data to explain the low percentage of CAR-NK cells in peripheral blood in this animal model.

References:

1. Tian Z, Chen Y, Gao B. Natural killer cells in liver disease. *Hepatology* **57**, 1654-1662 (2013).
2. Liu P, Chen L, Zhang H. Natural Killer Cells in Liver Disease and Hepatocellular Carcinoma and the NK Cell-Based Immunotherapy. *J Immunol Res* **2018**, 1206737 (2018).
3. Sun H, Sun C, Tian Z, Xiao W. NK cells in immunotolerant organs. *Cell Mol Immunol* **10**, 202-212 (2013).
4. Dimitrov S, Lange T, Born J. Selective mobilization of cytotoxic leukocytes by epinephrine. *J Immunol* **184**, 503-511 (2010).
5. Bogdanos DP, Gao B, Gershwin ME. Liver immunology. *Compr Physiol* **3**, 567-598 (2013).